# StealthInk: A Multi-bit and Stealthy Watermark for Large Language Models

**Ya Jiang** [1 2]   **Chuxiong Wu** [1]   **Massieh Kordi Boroujeny** [2]   **Brian Mark** [2]   **Kai Zeng** [2]

## Abstract

Watermarking for large language models (LLMs) offers a promising approach to identifying AI-generated text. Existing approaches, however, either compromise the distribution of original generated text by LLMs or are limited to embedding zero-bit information that only allows for watermark detection but ignores identification. We propose *StealthInk*, a stealthy multi-bit watermarking scheme that preserves the original text distribution while enabling the embedding of provenance information, such as userID, TimeStamp, and modelID, within LLM-generated text. This enhances fast traceability without requiring access to the language model's API or prompts. We derive a lower bound on the number of tokens necessary for watermark detection at a fixed equal error rate, which provides insights on how to enhance the capacity. Comprehensive empirical evaluations across diverse tasks highlight the stealthiness, detectability, and resilience of StealthInk, establishing it as an effective solution for LLM watermarking applications.

## 1. Introduction

Large language models (LLMs) like ChatGPT (OpenAI, 2022) or the open-sourced LLaMA (Touvron et al., 2023) and Gemini (Team et al., 2023), which have the remarkable ability to generate high-quality text, have become a crucial part of various text generation APIs, such as question answering, blog creation, and programming assistance (Austin et al., 2021; Perkins, 2023). However, these sophisticated language models' increased availability and capabilities present a significant concern. A key concern is their potential for facilitating the creation of fake news, which could be disseminated at scale, influencing public opinion and democratic processes more easily. Moreover, there is a risk of misuse of LLMs for scams or academic plagiarism. To tackle these issues, it is crucial to implement regulations and technical protections to encourage fair and responsible usage. Major providers of LLMs, including OpenAI, Google, and Meta, have committed to watermarking text generated by their models as part of this effort (Bartz & Hu., 2023).

Several papers suggest incorporating invisible watermarks into text to detect AI-generated content (Kirchenbauer et al., 2023; 2024; Christ et al., 2024; Kuditipudi et al., 2024; Hu et al., 2024; Wang et al., 2024; Zhao et al., 2024; Aaronson, 2023; Wu et al., 2024). However, detecting whether text is watermarked is insufficient to prevent misuse by malicious users.

Rather than solely embedding a zero-bit watermark to identify AI-generated content, it is crucial to employ a multi-bit watermark to track text provenance. Previous multi-bit watermarking schemes can be categorized into two groups: 1) undetectable or distortion-free watermarking, in which the watermarked text has exactly the same distribution as the non-watermarked text, e.g., (Kordi Boroujeny et al., 2024) and (Zamir, 2024), and 2) watermarking that allows some distortion of the output distribution (Wang et al., 2024; Fernandez et al., 2023; Yoo et al., 2024; Qu et al., 2024). A notable issue not tackled by the schemes in the first group is robustness against modifications made to the text generated by these methods.

In the second group, (Wang et al., 2024) use a proxy language model to guide LLM text generation according to the message, requiring extra resources and distorting the LLM's distribution. Other works (Fernandez et al., 2023; Qu et al., 2024) cyclically shift vocabulary permutations according to the message and bias tokens in a green list to enable efficient multi-bit decoding. However, overlapping shifts introduce interference, weakening message distinctiveness and statistical separation. To counteract this, a stronger bias is needed for reliable extraction, but this increases text distortion and reduces stealthiness. The multi-bit watermarking schemes proposed by (Yoo et al., 2024) essentially extend the reweighting approach outlined in (Kirchenbauer et al., 2023). This approach biases the frequency of certain words or phrases to embed the watermark during the generation process, such that the output distribution deviates from the

---

[1]Department of Computer Science, George Mason University, Fairfax, VA, USA [2]Wireless Cyber Center, College of Engineering and Computing, George Mason University, Fairfax, VA, USA. Correspondence to: Ya Jiang <yjiang25@gmu.edu>.

*Proceedings of the $42^{nd}$ International Conference on Machine Learning*, Vancouver, Canada. PMLR 267, 2025. Copyright 2025 by the author(s).

distribution of the LLM. Thus, the naturalness and readability of the LLM output may be disrupted, reducing the utility and effectiveness of the LLM for the users. Furthermore, due to lack of stealthiness, these schemes (Fernandez et al., 2023; Qu et al., 2024; Yoo et al., 2024) are vulnerable to the watermark spoofing attack (Jovanović et al., 2024) whereby an attacker forges a text watermarked by the LLM at low cost without knowing the secret key of the LLM by analyzing the distribution of n-grams between watermarked and non-watermarked texts. This poses a significant security risk, as malicious actors could fabricate harmful content that appears to be verified by a trusted model, thereby compromising the credibility of watermark-based verification systems. In contrast, a stealthy or undetectable watermark significantly increases the difficulty for adversaries attempting to manipulate or forge watermarked text, enhancing the resilience of the system against spoofing attacks and strengthens the reliability of watermark-based authentication.

We propose a novel reweighting strategy for multi-bit watermarking that satisfies all of the properties listed in Table 1, which compares our proposed *StealthInk* scheme with SOTA multi-bit watermarking schemes. The stealthiness property implies that the watermarked text is statistically difficult to distinguish from non-watermarked text without the watermark key. Efficiency refers to the low computational complexity of the encoder and decoder. High AUC implies low false positive and false negative rates in detecting the watermark. For multi-bit watermarking, high bit accuracy in decoding the hidden message is an important property. In Section 5, we provide a theoretical derivation of the minimum number of tokens necessary for watermark detection at a fixed equal error rate, i.e., the false positive and false negative rates are equal, which provides insights for capacity enhancement.

*Table 1.* Comparison with SOTA multi-bit LLM watermarking schemes.
**A**: (Kordi Boroujeny et al., 2024; Zamir, 2024); **B:** (Wang et al., 2024; Fernandez et al., 2023); **C:** (Yoo et al., 2024; Qu et al., 2024).

|                 | A | B | C | StealthInk |
|-----------------|---|---|---|------------|
| Stealthiness    | ✓ | ✗ | ✗ | ✓          |
| Efficiency      | ✓ | ✗ | ✓ | ✓          |
| High AUC        | ✓ | ✓ | ✓ | ✓          |
| High bit accuracy | ✓ | ✓ | ✓ | ✓        |
| Robustness      | ✗ | ✓ | ✓ | ✓          |

We empirically compare StealthInk with existing methods (Yoo et al., 2024; Qu et al., 2024; Fernandez et al., 2023), demonstrating its superiority in detection, extraction, text quality, and stealthiness. StealthInk achieves an AUC of 0.98 and a bit accuracy of 0.92 when embedding 24-bit messages in 300 tokens. In contrast, prior methods either degrade text quality for better detectability (Yoo et al., 2024; Qu et al., 2024) or sacrifice detectability to maintain qual-

ity (Fernandez et al., 2023), while also lacking stealthiness and being vulnerable to spoofing. StealthInk ensures robust detection, high text quality, and enhanced security, making it resilient against attacks.

## 2. Preliminaries

**Notation:** We first establish a few essential notations. Let $V$ denote the vocabulary set and $|V|$ its size. We use $\boldsymbol{x}_{1:L}$ to denote the sequence of $L$ tokens $\{\boldsymbol{x}_1, \ldots, \boldsymbol{x}_L\}$; $\boldsymbol{x}_{:i}$ denotes the first $i$ elements while $\boldsymbol{x}_{-i:}$ represents the last $i$ elements. Denote the prompt as $\boldsymbol{a}$ and the likelihood of generating an imminent token $\boldsymbol{x}_{L+1} \in V$ given the prompt and current context by $P_O(\boldsymbol{x}_{L+1} \mid \boldsymbol{a}, \boldsymbol{x}_{1:L})$. Due to the autoregressive model of operation that an LLM adopts, the joint probability of producing $L$ tokens spanning from $\boldsymbol{x}_1$ to $\boldsymbol{x}_L$ is given by

$$P_O(\boldsymbol{x}_{1:L} \mid \boldsymbol{a}) = \prod_{i=1}^{L} P_O(\boldsymbol{x}_i \mid \boldsymbol{a}, \boldsymbol{x}_{1:i-1}).$$

Like other LLM watermarking schemes, our approach utilizes pseudorandom functions (PRFs) (Goldreich et al., 1986) to generate an i.i.d. watermark cipher $\theta$ with a uniform distribution. Specifically, $\theta$ is a permutation of the vocabulary $V$, derived by seeding the PRF with the watermark key $\mathsf{sk} \in \mathcal{SK}$ and the texture key $s$, i.e., the context n-grams $\boldsymbol{x}_{-h:} \in V^h$. This results in an ordered sequence of tokens $\theta = (t_1, \ldots, t_k, \ldots, t_{|V|})$, where $t_k$ is the $k$th token in the permutation. We denote an $m$-bit watermark message as $M \in \mathcal{M} = \{0, 1, \ldots, 2^m - 1\}$. In addition, we define $P_W^M(\boldsymbol{x}_{L+1} \mid \boldsymbol{a}, \boldsymbol{x}_{1:L}, \theta_L)$ as the watermarked probability distribution for generating the $(L+1)$-th token.

## 3. Threat model

Following (Jovanović et al., 2024), the model owner deploys an LLM $LM_{mo}$ with a watermarking scheme and a watermark key $\mathsf{sk}$. We assume the attacker has only black-box access (e.g., API access) to full generations of $LM_{mo}$ and is aware of the presence of the watermark behind the API. In line with standard security assumptions (Kerckhoffs' principle), we assume that the attacker knows all parameters of the watermarking scheme, but not $\mathsf{sk}$.

**Editing attacks** To avoid watermark detection, some attackers could try to edit the text while ensuring text quality. In that case, after obtaining the response from the LLM, attackers could employ straightforward text modification, such as substitution, insertion, and deletion, while ensuring the preservation of semantics. This method aims to subtly alter the content to evade detection, making it challenging for conventional security measures to identify the modified text. Distinct from modification attacks, in paraphrasing attacks, the attackers focus on rephrasing the text rather than making

direct modifications. This involves rearranging sentence structures, changing the order of words, or altering grammatical constructions to achieve a different phrasing while preserving the underlying semantics. The intent is to create a response that is different but retains the core meaning, which complicates the task of identifying the manipulated content.

**Watermark spoofing attacks.** Targeting distribution-modified watermarking schemes that boost green-list tokens by adding $\delta$ to their logits (Kirchenbauer et al., 2023; 2024; Zhao et al., 2024; Yoo et al., 2024; Qu et al., 2024), Jovanovic et al. (Jovanović et al., 2024) introduce watermark spoofing. Here, an attacker queries $LM_{mo}$ minimally to approximate its watermarking rules based on sk, enabling the forgery of text detected as watermarked, thus damaging the LLM provider's reputation. This attack assumes access to an auxiliary model $LM_{att}$—a trivial condition given the abundance of open models.

(Jovanović et al., 2024) spoof non-stealthy zero-bit watermarks by identifying distribution differences between watermarked and non-watermarked text. For instance, (Kirchenbauer et al., 2023) boost logits of tokens in the first half of a permutation, increasing their selection likelihood. As a result, certain tokens after context n-grams $\boldsymbol{x}_{-h:}$ consistently have higher probabilities than in a non-watermarked LLM.

However, multi-bit watermarks embed diverse signals across tokens. When $m = 1$, some tokens carry signal 0, others signal 1, creating varying bias directions. This additional complexity makes spoofing multi-bit watermarks more challenging. In Section 6, we investigate the potential to extend the stealing attack proposed by (Jovanović et al., 2024) to multi-bit watermarking schemes.

# 4. Methodology

We propose a multi-bit and stealthy watermarking scheme, *StealthInk*, which uses a novel token sampling probability reweighting strategy to embed information in the sampled tokens from randomized vocabulary permutations. We double the probabilities of certain tokens while ensuring that several others have a probability of zero, thus preserving the expected distribution over the random space of permutation. The tokens with zero probability, referred to as *red tokens*, are never sampled, enabling watermark detection and information bit extraction through statistical tests. In this section, we will first extend the stealthy concept from zero-bit watermarking to multi-bit watermarking, and then describe StealthInk in detail.

## 4.1. Stealthy multi-bit watermarking

We adopt the definition of a stealthy or unbiased watermark used in (Hu et al., 2024; Wu et al., 2024) and extend the concept to multi-bit watermarking. Let $\Theta$ denote

a set of watermark ciphers and let $P_\Theta$ denote a probability distribution defined on $\Theta$. We assume that $P_\Theta$ is the uniform distribution on $\Theta$ since $\theta \in \Theta$ is determined by a PRF. Let $\Delta_V$ denote the set of all possible probability distributions on $V$. We then define reweighting function $F(\theta, M, P_O) : \Theta \times \mathcal{M} \times \Delta_V \to \mathbb{R}^{|V|+1}$, which generates the watermarked text with embedded message $M$.

**Definition 4.1.** (Stealthy or unbiased multi-bit watermark) After embedding $M \in \mathcal{M}$, the watermarked distribution, $P_W^M$, determined by the reweighting function $F(\theta, M, P_O)$ is *stealthy* or *unbiased* if for all $P_O \in \Delta_V$, $\mathbb{E}_{\theta \sim P_\Theta}[P_W^M] = P_O$, where $P_W^M = G(F(\theta, M, P_O)) \in \Delta_V$ and $G$ is a difference operator that transforms a cumulative vector into a discrete probability distribution over the vocabulary.

Definition 4.1 establishes the unbiasedness of the watermarked distribution for a single generated token. We further define the unbiasedness of the watermarked distribution for texts. Note that for a single response where $M$ is given and fixed, stealthiness or unbiasedness is not relevant, as a user cannot infer a probability distribution from just one output. The concept only becomes meaningful when considering multiple responses over time, where the message changes and is random due to metadata variations (e.g., userID, TimeStamp).

In the following, to define the stealthiness over $K$ pairs of prompt and response, we use bold lowercase $\boldsymbol{a}^k$ for the $k$th prompt and $\boldsymbol{x}^k = (x_1^k, \ldots, x_{L^k}^k)$ to denote a response of a sequence of $L^k$ tokens in the $k$th response. The embedded message in the $k$th response is denoted $M^k$, and $\boldsymbol{\theta}^k = (\theta_1^k, \ldots, \theta_{L^k}^k)$ represents the sequence of vocabulary permutations used during generation. The watermarked probability distribution for generating the entire text $\boldsymbol{x}^k$ is denoted as $\mathbb{P}_{wm}(\boldsymbol{x}^k \mid \cdot)$.

**Definition 4.2** ($K$-shot stealthy watermark). Let $\{\boldsymbol{a}^k, \boldsymbol{x}^k\}_{k=1}^K$ be $K$ prompt–response pairs, with aforementioned notations, the watermarking scheme is said to be $K$-shot stealthy if the expectation of joint distribution over the $K$ generations under watermarking matches the original model's joint distribution:

$$\mathbb{E}_{\boldsymbol{\theta}^{1:K}, M^{1:K}}[\mathcal{A}] = \prod_{k=1}^K P_O(\boldsymbol{x}^k \mid \boldsymbol{a}^k), \qquad (1)$$

where

$$\mathcal{A} = \prod_{k=1}^K \mathbb{P}_{wm}\left(\boldsymbol{x}^k \mid \boldsymbol{a}^k, M^k, \boldsymbol{\theta}^k, \boldsymbol{a}^{1:k-1}, \boldsymbol{x}^{1:k-1}, M^{1:k-1}\right).$$

The following theorem illustrates the condition when $K$-shot stealthiness holds. See the proof in Appendix B.

**Theorem 4.3.** *For any prompt $\boldsymbol{a}^k$ and the response $\boldsymbol{x}^k$, if the sequence of ciphers $\boldsymbol{\theta}^k$ is independent, and each token*

*is generated with unbiasedness, then the $K$ pairs of prompts $\boldsymbol{a}^{1:K}$ and responses $\boldsymbol{x}^{1:K}$ are $K$-shot stealthy.*

If two seeds applied to the PRF are different then the two generated watermark ciphers, say $\boldsymbol{\theta}_i^k$ and $\boldsymbol{\theta}_j^k$ ($i, j \in [1, L^k], i \neq j$), will "look" random and independent. To ensure the independence of $\boldsymbol{\theta}^k$, intuitively, when the same seed is encountered during generation of $\boldsymbol{x}^k$, the original distribution $P_O(\boldsymbol{x}_i^k | \boldsymbol{a}^k, \boldsymbol{x}_{1:i-1}^k)$ should be used. Otherwise, the reweighted probability distribution $P_W^{M^k}(\boldsymbol{x}_i^k | \boldsymbol{a}^k, \boldsymbol{x}_{1:i-1}^k, \boldsymbol{\theta}_i^k)$ is applied. Therefore, the probability distribution of generated tokens either follows the original distribution or an unbiased distribution. In that case, according to Theorem 4.3, the watermarked probability distribution of multiple pairs of prompts and responses can be preserved.

The property of $K$-shot stealthiness ensures that the watermark remains imperceptible to both users and potential attackers, even if they are aware of the original distribution $P_O$. Specifically, it makes watermark spoofing attacks more difficult for attackers to successfully execute by distinguishing between the watermarked text distribution and the non-watermarked text distribution, as evaluated in Section 6. Moreover, preserving the text distribution inherently implies maintaining the quality of the generated text.

### 4.2. StealthInk

**Notation:** We define $\gamma_M = M \cdot \gamma$, where $\gamma = 2^{-m}$, to map $M$ to an interval in a permutation $\theta$. Consequently, $\gamma_M \in \{0, 2^{-m}, 2^{1-m}, \ldots, 1 - 2^{-m}\}$. Tokens within the interval associated with $M$ are indexed as[1] $\theta[\gamma_M |V| : (\gamma_M + \gamma)|V|]$. To generate the $i$th token, given the original probability distribution $P_O^i$ and the permutation $\theta_i$ seeded by $\boldsymbol{x}_{-h:}$, an axis ranging from 0 to 1 can be constructed. Along this axis, the tokens are arranged in the order defined by $\theta_i$, occupying positions proportional to their respective probabilities. By denoting $X_k = \sum_{j=1}^{k} P_O^i(t_k | \boldsymbol{a}, \boldsymbol{x}_{1:i-1}, \theta_i)$, we can define $X_{k-1}$ and $X_k$ as the left and right cumulative probabilities corresponding to token $t_k$, which is the $k$th token in $\theta_i$. Then the probability of $t_k$ under $P_O^i$ is $X_k - X_{k-1}$.

Next, define $\alpha$ to be sum of the probabilities of the tokens $t_1, t_2, \ldots, t_{\gamma_M |V|}$, under probability distribution $P_O^i$ with permutation $\theta_i$:

$$\alpha = \sum_{j=1}^{\gamma_M |V|} P_O^i(t_j | \boldsymbol{a}, \boldsymbol{x}_{1:i-1}, \boldsymbol{\theta}_i) = X_{\gamma_M |V|}, \quad (2)$$

i.e., $\alpha$ is the left cumulative probability corresponding to the

---

[1]With slight abuse of notation, for a permutation $\theta = (t_1, \ldots, t_k, \ldots, t_{|V|})$, we denote the subsequence $(t_a, \ldots, t_b)$ as $\theta[a : b]$ and we assume that the ceiling operation is applied to indices, e.g., as an index, $\gamma_M |V|$ is interpreted as $\lceil \gamma_M |V| \rceil$.

$\gamma_M |V|$-th token. Similarly, define

$$\beta = \sum_{j=1}^{(\gamma_M + \gamma)|V|} P_O^i(t_j | \boldsymbol{a}, \boldsymbol{x}_{1:i-1}, \boldsymbol{\theta}_i) = X_{(\gamma_M + \gamma)|V|}, \quad (3)$$

i.e., $\beta$ is the right cumulative probability corresponding to the $(\gamma_M + \gamma)|V|$-th token. Let $\bar{\beta} = 1 - \beta$ and $\bar{\alpha} = 1 - \alpha$.

Fig. 1 presents a simplified example of a StealthInk encoder for embedding a message of 8 bits. The message is formed as a $MsgSeq$, which is comprised of $H$ message chunks and we refer to the index of the chunk as $pos \in \{0, 1, \ldots, H - 1\}$. Each message chunk carries an $m$-bit signal. Therefore, $MsgSeq$ carries an $Hm$-bit signal. In Fig. 1, $m = 2$, $H = 4$, and $\gamma = 2^{-m} = 0.25$. To enhance the capacity, we leverage the position allocation approach in MPAC, which allocates each token pseudorandomly onto a message chunk of $MsgSeq$ to be embedded.

As illustrated in Fig. 1, given a prompt, StealthInk first derives the texture key using the last n-grams of the prompt (in this case, $h = 3$). The watermark key and texture key are then combined to form a seed for a PRF, which generates a permutation over the vocabulary set $V$. This seed is also used to choose a position $pos$, determining the message chunk to embed at the current generation step.

In Fig. 1, $pos = 1$, meaning that the corresponding message chunk "10" (i.e., $M = 1$) is selected for embedding at this step. By forming the permutation with each token ordered according to their probabilities from $P_O$, we compute $\alpha$ and $\beta$ as defined in (2) and (3). This process allows us to define the red list with $M$ embedded,

$$RL^M = \theta[\gamma_M |V| : (\gamma_M + \gamma)|V|], \quad (4)$$

where tokens within this range have their probabilities reweighted to zero and will not be sampled.

To demonstrate how StealthInk works, we present four cases in Fig. 2. In each case, $2\star$ indicates that the probabilities of tokens in that interval are doubled, while 0 means the probabilities of tokens in that interval are zeroed. In addition, $\star$ indicates that the probabilities of tokens in the corresponding interval keep their original values. When embedding the message $M$, the probabilities of tokens in the red list will be reweighted to 0. Let $F(\theta, M, P_O)_k$ be the reweighted cumulative probability for the token $t_k$ under permutation $\theta$ for $k = 1, \ldots, |V|$, illustrated in Fig. 2, which can be expressed as[2]:

$$F(\theta, M, P_O)_k = \begin{cases} (X_k - \beta)^+ + (X_k - \bar{\beta})^+ \\ -(X_k - \alpha)^- - (X_k - \bar{\alpha})^+, & \text{Case 1 or 3,} \\ (X_k - \beta)^+ + (X_k - \bar{\beta})^- \\ -(X_k - \alpha)^- - (X_k - \bar{\alpha})^-, & \text{Case 2 or 4.} \end{cases}$$

(5)

---

[2]$(x)^+ := \max(x, 0)$ and $(x)^- := -\min(x, 0)$. Note that $x = (x)^+ - (x)^-$.

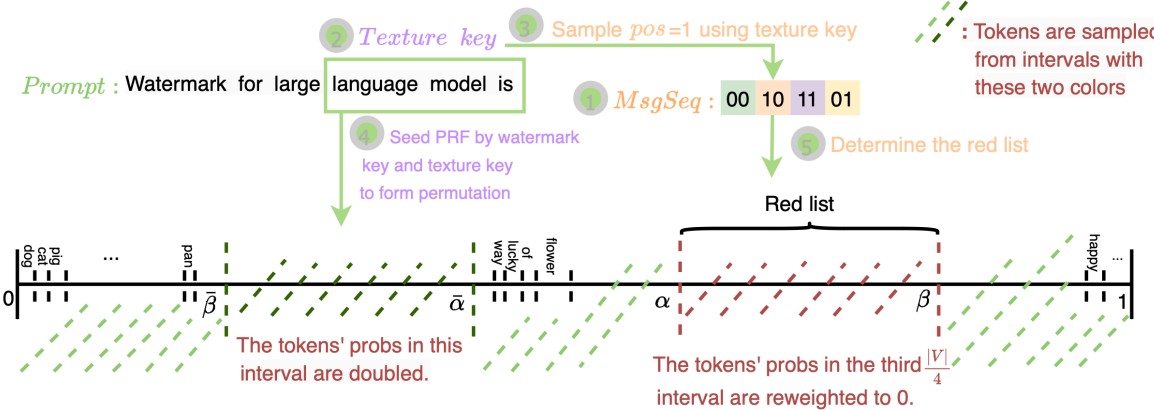

*Figure 1.* StealthInk encoder: $m = 2$, $H = 4$.

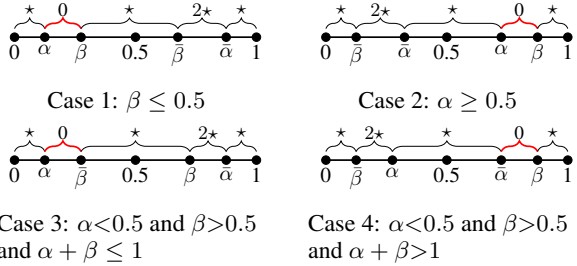

Case 1: $\beta \leq 0.5$    Case 2: $\alpha \geq 0.5$

Case 3: $\alpha < 0.5$ and $\beta > 0.5$ and $\alpha + \beta \leq 1$

Case 4: $\alpha < 0.5$ and $\beta > 0.5$ and $\alpha + \beta > 1$

*Figure 2.* Multi-bit watermarking reweighting rule.

With the original distribution $P_O^i$ at the $i$th generation step, the reweighted probability of token $t_k$ is $P_W^M(t_k | \boldsymbol{a}, \boldsymbol{x}_{1:i-1}, \boldsymbol{\theta}_i) = G(F(\theta, M, P_O^i))_k = F(\boldsymbol{\theta}_i, M, P_O^i)_k - F(\boldsymbol{\theta}_i, M, P_O^i)_{k-1}$. We prove the following result in Appendix C.

**Theorem 4.4.** *For each $\theta \in$ i.i.d. $\theta_{1:L}$, the reweighting function $F_k(\theta, M, P_O)$ in (5) ensures stealthy watermarking.*

Algorithm 1 shows the process of encoding a multi-bit watermark in StealthInk. During a query attempt, to ensure the independence between permutations, the same texture key should not repeat for watermarking, which means the original probability distribution is applied when encountering the same texture key. In this way, a history log hist is maintained for each query attempt to record the texture key encountered.

In Dipmark (Wu et al., 2024) and ZMH (Hu et al., 2024) (see Algorithm 1 in both), a single hist must be shared across all user queries to maintain $K$-shot stealthiness. However, as global queries increase, the pool of available texture keys shrinks, requiring periodic replacement of the watermark key sk. This necessitates keeping track of all used keys, which adds overhead in both storage and verification.

This issue becomes even more prominent when embedding the multi-bit watermark using only multiple keys or en-

hancing StealthInk's capacity by key enumeration. For instance, embedding a 2-bit message requires choosing from 4 watermark keys. As shown in Appendix L, traversing sk values causes significant detection delays without necessarily improving capacity. Furthermore, since StealthInk leverages the position allocation approach in MPAC (Yoo et al., 2024) to enhance capacity, ensuring the randomness of the message is crucial for maintaining $K$-shot stealthiness. To achieve this, the lower-end bits of TimeStamp, such as those representing seconds or even milliseconds, should be embedded in the initial tokens of the response. The remaining message chunks are then embedded into the rest of the tokens using position allocation. Therefore, even if a user repeatedly queries the same prompt, the distribution of the responses is preserved. Moreover, an attacker attempting a spoofing attack by distinguishing the distributions of the original and watermarked text can only succeed if they use the same userID to repeatedly submit the same prompt within a very short time frame (e.g., within 1 second or even 1 millisecond) and gather a sufficiently large number of queries (e.g., 100,000). This constraint makes it significantly more challenging for an attacker to exploit distributional differences to forge the watermark. In Appendix F, we further discuss resilience against adaptive attacks through a sufficiently large number of queries.

In the detection stage, as described in Algorithm 2 in Appendix D, each token in the given text is mapped to a position $pos$ by seeding a PRF with the watermark and texture keys. For tokens mapped to a particular $pos$, we count the number of tokens that fall into the red list defined by $M \in \mathcal{M}$, as specified in Eq. (4). Under the null hypothesis (i.e., the text is not watermarked), the probability that a token falls into the red list is $\gamma$. Let $L_{pos}$ denote the number of tokens mapped to position $pos$; then the number of red list tokens $R_{pos}^M \sim \text{Bin}(L_{pos}, \gamma)$. We extract the message chunk at each position by selecting the candidate $M \in \mathcal{M}$ that minimizes the count of red list tokens

$M^*_{pos} = \arg\min_{M \in \mathcal{M}} R^M_{pos}$. To detect whether the text is watermarked, we concatenate $M^*_{pos}$ across all positions $pos \in \{0, 1, \ldots, H-1\}$ to form the message sequence $MsgSeq$. Then, we compute the $z$-score as

$$Z(MsgSeq, x_{1:L}) = \frac{R - \hat{\mu}_R}{\hat{\sigma}_R}, \qquad (6)$$

where $\hat{\mu}_R$ and $\hat{\sigma}_R$ denote the empirical mean and standard deviation of $R$, estimated from a collection of non-watermarked texts. Note that in our experiments, we evaluate detection performance by comparing the values of $R$ for watermarked and non-watermarked texts, rather than setting a threshold on the $z$-score.

---

**Algorithm 1** StealthInk Encoder

---

**Input:** $\boldsymbol{a}$, sk, $L$, $h$, $f$, hist $= \emptyset$, $MsgSeq$, $H$, $\gamma = 2^{-m}$

  **for** $i = 1, \ldots, L$ **do**
    Calculate $P_O(\cdot \,|\boldsymbol{a}, \boldsymbol{x}_{:i-1})$
    Generate texture key $s_i$ based on $\boldsymbol{x}_{-h:}$
    Sample $pos$ from $[0, 1, \ldots, H-1]$ with seed $s_i$; let $MsgSeq[pos] = M$ and $\gamma_M = M\gamma$
    **if** $s_i \in$ hist **then**
      Sample $\boldsymbol{x}_i$ from $P_O(\cdot \,|\boldsymbol{a}, \boldsymbol{x}_{:i-1})$
    **else**
      hist.add$(s_i)$
      Generate permutation $\boldsymbol{\theta}_i = f(\text{sk}, s_i)$
      Calculate $\alpha$ and $\beta$ using (2) and (3)
      Calculate $P_W^M(\cdot \,|\boldsymbol{a}, \boldsymbol{x}_{:i-1}, \boldsymbol{\theta}_i)$ from (5)
      Sample $\boldsymbol{x}_i$ from $P_W^M(\cdot \,|\boldsymbol{a}, \boldsymbol{x}_{:i-1})$
    **end if**
  **end for**

---

## 5. Theoretical analysis

In Appendix G we can derive the probability of sampling a red token when embedding the message $M^*$ as

$$\mathbb{P}[rt^{\gamma_{M^*}}] = \frac{(1-p)(\beta - \alpha)}{2^{m+1}} + p(\beta - \alpha) \qquad (7)$$

where $p$ represents the probability of repeated texture keys, defined as the ratio of the number of texture keys that have already appeared to the total number of texture keys in a text. As $\mathbb{P}[rt^{\gamma_{M^*}}]$ increases, under a certain equal error rate (eer), the minimum number of tokens required to detect the watermark $L_{\min}$ is larger. Specifically, we present the relationship between $L_{\min}$ and eer under uniform distribution as Fig. 3, which is the best $L_{\min}$ with the maximum entropy. In that case, $\beta - \alpha = 2^{-m}$.

One key insight is that capacity can be enhanced by distributing unit capacity across locations. As shown in Fig. 3, the $L_{\min}$ needed to embed 2 bits is smaller than half of that required for 4 bits. Thus, embedding 2 bits into two separate

chunks can achieve the same eer with significantly fewer tokens. By analyzing the relationship between $L_{\min}$ and eer under varying entropy and repetition rates $p$, we can determine the optimal unit capacity for improving capacity through position allocation.

Another insight from Fig. 3 is that repetitions greatly hinder watermark detection. When $p = 0.2$, the minimum number of tokens required for detection is much higher than when $p = 0$. As discussed in Section 4.2, Dipmark and ZMH share a global history log across all queries, causing $p$ to increase over time, requiring more tokens for watermarking. In contrast, our method constructs an empty history log for each query and recycles it after response completion, which keeps $p$ significantly lower and stabilizes $L_{\min}$ across different query attempts through our reweighting strategy.

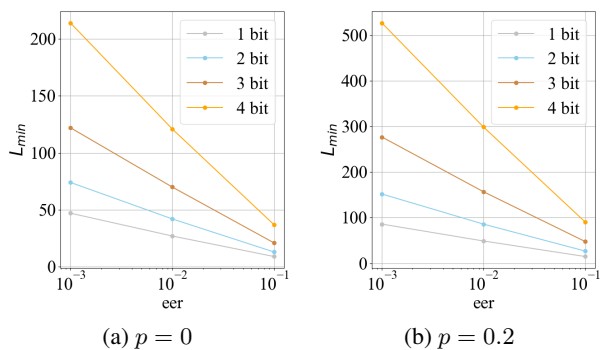

(a) $p = 0$            (b) $p = 0.2$

*Figure 3.* $L_{\min}$ vs. eer for $p = 0$ and $p = 0.2$.

## 6. Experiments

We compare StealthInk with SOTA methods (Yoo et al., 2024; Qu et al., 2024; Fernandez et al., 2023) on stealthiness, detectability, and robustness. For text completion, unless noted otherwise, we use LLAMA2-7B (Touvron et al., 2023) and 500 randomly selected texts from the RealNewsLike subset of C4 (Raffel et al., 2020), trimming a fixed number of tokens from the start as prompts (see Appendix H).

### 6.1. Stealthiness Results

#### 6.1.1. STEALTHINESS PRESERVES TEXT QUALITY

Following Dipmark and ZMH, we compare text quality among the methods for machine translation, text summarization, and text completion tasks. For the text summarization task, we employ the BART-large model (Liu et al., 2020). For the machine translation task, we focus on English-to-Romanian translation and employ the Multilingual BART (MBart) model (Liu et al., 2020) on the WMT'14 En-Ro corpus (Bojar et al., 2014). Due to the limited number of tokens in each sample, which is less than 50 tokens, we compare the quality of text watermarked by 1 bit and 2 bit watermarks, respectively, for machine translation and text summarization tasks, as shown in Table 2, where no segmen-

*Table 2.* Text quality for machine translation and text summarization. PPL is the median perplexity.

| Metrics | Machine Translation | | | | | | Text Summarization | | | |
|---|---|---|---|---|---|---|---|---|---|---|
| | BERTScore ↑ | | BLEU ↑ | | ROUGE-1 ↑ | | PPL ↓ | | ROUGE-1 ↑ | |
| Unit Capacity (Bit) | 1 | 2 | 1 | 2 | 1 | 2 | 1 | 2 | 1 | 2 |
| Non-Watermarked | 0.6995 | | 0.0677 | | 0.3312 | | 7.274 | | 0.2319 | |
| StealthInk | **0.6941** | **0.6967** | **0.0776** | **0.0719** | **0.3317** | **0.3337** | **7.147** | **7.333** | **0.2322** | **0.2325** |
| MPAC | 0.6457 | 0.6515 | 0.0442 | 0.0488 | 0.2670 | 0.2789 | 10.51 | 10.82 | 0.1991 | 0.1963 |
| Qu et al. | 0.5840 | 0.5822 | 0.0180 | 0.0153 | 0.1782 | 0.1628 | 26.27 | 25.09 | 0.1498 | 0.1512 |
| Fernandez et al. | 0.6658 | 0.6730 | 0.0489 | 0.0483 | 0.2807 | 0.2907 | 12.41 | 12.58 | 0.2083 | 0.2094 |

*Table 3.* Spoofing attacks on different schemes with 1 bit and 2 bits embedded, respectively, where $m = 1$.

| | | Dolly CW | | | | MMWBookReports | | | |
|---|---|---|---|---|---|---|---|---|---|
| | | AUC ↓ | TPR@10%FPR ↓ | @FNR*1e-3 ↓ | GPT4 ↑ | AUC ↓ | TPR@10%FPR ↓ | @FNR*1e-3 ↓ | GPT4 ↑ |
| $H$=1 | StealthInk | **0.7265** | **0.29** | **0.4482** | 8.11 | **0.8064** | **0.5** | **0.3333** | **8.74** |
| | MPAC | 0.8301 | 0.6 | 0.7018 | **8.22** | 0.86 | 0.73 | 0.6885 | 8.24 |
| | Qu et al. | 0.9835 | 0.97 | 0.9899 | 6.77 | 0.954 | 0.93 | 1.0 | 7.0 |
| $H$=2 | StealthInk | **0.8456** | **0.57** | **0.6667** | **8.73** | **0.893** | **0.65** | **0.5439** | **8.92** |
| | MPAC | 0.9239 | 0.83 | 0.8313 | 8.21 | 09491 | 0.84 | 0.7125 | 8.67 |
| | Qu et al. | 0.9979 | 1.0 | 1.0 | 7.0 | 0.999 | 1.0 | 1.0 | 7.1 |

tation is utilized, i.e., $H = 1$. On the other hand, for text completion, we present the quality of text watermarked by 24 bits at 200 tokens for different schemes in Fig. 6, where segmentation is leveraged, $m = 1$ and $H = 24$.

As shown in Table 2 and Fig. 6 in Appendix I, text quality remains largely unaffected by watermark capacity, regardless of whether it is 1 bit, 2 bits, 24 bits, or more. Both Table 2 and Fig. 6 shows that text quality remains consistent between StealthInk watermarked and non-watermarked texts. In contrast, the other methods degrade text quality.

### 6.1.2. STEALTHINESS PROTECTS AGAINST SPOOFING

Following the settings in (Jovanović et al., 2024), we compare the robustness against watermark spoofing attack by setting up $LM_{mo}$ as LLAMA2-7B and $LM_{att}$ as MISTRAL-7B (Jiang et al., 2023). We randomly select 30,000 prompts from the RealNewsLike subset of the C4 dataset to query each LLM, generating responses of fewer than 800 tokens under various watermarking schemes, in order to learn and estimate the underlying watermark patterns. $LM_{att}$ then generate 100 responses of less than 800 tokens, respectively, using the prompts from Dolly-CW (Conover et al., 2023) and MMW BookReports (Piet et al., 2023). Table 3 presents the performance of spoofing attacks on different multi-bit watermarking schemes for embedding 1 bit ($H = 1$) and 2 bits ($H = 2$), respectively, where $m = 1$. We exclude Fernandez et al. in this comparison because of its low detectability. In particular, we ignore low-quality texts (GPT4 score below 6.5). From the attacker's view, the forged text should be detected as watermarked, which is a positive sample, while the true non-watermarked text which is a negative sample should be correctly identified as well. AUC is de-

rived to test the performance of watermarking schemes in distinguishing between watermarked and non-watermarked texts. Low AUC would indicate a failed spoofing attack, which means the detector cannot discern the watermarked text forged by the attacker from the non-watermarked text. TPR@10%FPR represents the fraction of forged texts that are detected as watermarked when the ratio of false identified non-watermarked texts is smaller than 10%. @FNR*1e-3 denotes the fraction of forged texts that are detected as watermarked at FNR=1e-3 (i.e., p-value ≤ 0.001). Therefore, smaller TPR@10%FPR and @FNR*1e-3 represent an ineffective spoofing attack. We observe that the text forged from StealthInk consistently exhibits the lowest detection performance among the evaluated methods. For instance, when an attacker learns the outputs watermarked by StealthInk and attempts to forge watermarked text on the Dolly CW dataset, only 29% of the forged texts are detected as watermarked when the FPR is below 10%. In contrast, forged texts generated using MPAC and (Qu et al., 2024) achieve significantly higher detection rates, with about 70% and 98% of forgeries being correctly identified as watermarked under the FNR = $10^{-3}$ setting.

### 6.2. Detectability results

First, we compare the TPR for $m \in \{1, 2, 3, 4\}$ when the FPR is below 1%, with $H = 1$, to determine the optimal value of $m$ for constructing the $MsgSeq$, as shown in Fig. 4. Using the red dashed line, which represents the TPR for embedding 1 bit in 50 tokens, we observe that the TPR for embedding 2 bits in 100 tokens is lower, and the TPR for embedding 3 and 4 bits in 150 and 200 tokens, respectively, is even less than half. Therefore, we select $m = 1$.

Then we evaluate the performance of message detection and

extraction among various methods in 24 bits, 36 bits, and 48 bits at 200 tokens, as shown in Table 4. While the method of (Qu et al., 2024) achieves the best detection and extraction accuracy, their text quality measured by perplexity is significantly compromised. (Fernandez et al., 2023) exhibit lower TPR and bit accuracy with significantly higher time cost compared to other methods. For example, when embedding 24-bit messages, it cost 113 s to extract the message. In contrast, StealthInk and MPAC extract the message in only 0.01 s. This highlights the inefficiency of directly using varied keys within the hash function.

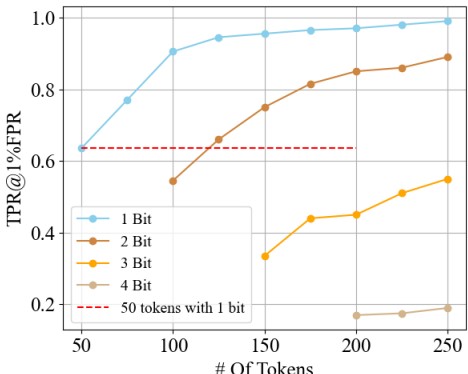

*Figure 4.* Detection performance comparisons with $m \in \{1, 2, 3, 4\}$ and $H = 1$.

In particular, we compare the TPR of StealthInk and MPAC with 1% FPR and 0.1%FPR, respectively, across various number of tokens in Fig. 5. The logits bias $\delta$ of MPAC are compared between 1, 1.5, and 2, where $\delta = 2$ is the original setting in MPAC and is applied in the evaluation of MPAC in this paper. Higher logits bias generally represents more skewed probability distribution and larger watermark detectability. We observe that while StealthInk may initially show lower TPR at shorter lengths (e.g., 200 tokens), its detectability approaches or matches MPAC as the number of tokens increases. For example, when embedding 36 bits, MPAC ($\delta = 2$) achieves TPR@ 1%FPR of 0.98 at 200 tokens, while StealthInk achieves comparable TPR@ 1%FPR (i.e., 0.985) at 400 tokens. Note that StealthInk's TPR is significantly better than that of MPAC when $\delta = 1$, while close to MPAC's TPR when $\delta = 1.5$. For example, when embedding 36 bits, MPAC ($\delta = 1.5$) achieves TPR@ 1%FPR of 0.97 at 300 tokens, while StealthInk achieves comparable TPR@ 1%FPR (i.e., 0.9725) at 400 tokens.

This demonstrates that StealthInk is competitive in detectability given sufficient sequence length, while offering additional benefits in terms of stealthiness, robustness to spoofing, and multi-bit capacity. Unlike distribution-modifying schemes like MPAC, which achieve higher detectability through aggressive token-level distortion, StealthInk is designed to be statistically stealthy across multiple

text generations with theoretical guarantees.

Note that in Table 4 the PPL of the StealthInk is significantly lower than the non-watermarked sequences, which arises from the evaluation setup and not from a violation of the stealthiness guarantee of StealthInk. StealthInk uses a reweighting strategy that satisfies two constraints during token sampling: (1) the token must not appear in the red list (whose probability is zeroed out), and (2) it must be sampled from the remaining tokens based on multinomial sampling. In contrast, non-watermarked text generation only follows the second condition. As a result, StealthInk effectively filters out low-probability tokens that fall into the red list, which may include semantically weak or low-quality options. This selection bias may cause the sampled tokens to have higher average probabilities, thus reducing PPL.

However, this observation does not contradict the stealthiness of StealthInk. The PPL reported in Table 4 are median values aggregated over responses of 200 tokens to 500 prompts, each associated with a random message and permutation. Due to this variability, small empirical differences in PPL can emerge across text generations. Nonetheless, StealthInk is provably stealthy in expectation (as shown in Definition 4.1 and Theorem 4.4), i.e., when averaged over a sufficiently large number of generated tokens or samples, the watermarked and non-watermarked token distributions are statistically indistinguishable. In Appendix J, we show the PPL with various number of generations and note that if we further increase the number of prompts or use even longer generated sequences, the PPL statistics of StealthInk and non-watermarked text become closer to each other.

## 6.3. Robustness against editing attacks

The copy-paste attack involves mixing watermarked text with non-watermarked text, while the paraphrasing attack rewrites the watermarked text using another language model to preserve its meaning. For the copy-paste attack, we randomly mix a proportion $\epsilon$ of non-watermarked text into the watermarked text, maintaining the total length. In the paraphrasing attack, we use the Dipper (Krishna et al., 2023) and PEGASUS (Zhang et al., 2020) models to paraphrase the watermarked text, with $\epsilon$ representing the paraphrasing strength.

As shown in Table 5, bit accuracy decreases steadily with the increasing proportion of non-watermarked text in the copy-paste attack. This occurs because replacing more watermarked text reduces the distinguishing features required for accurate watermark extraction. Additionally, non-watermarked tokens distribute the watermark signal more evenly, as seen in StealthInk, where non-watermarked tokens are spread across the vocabulary permutation. Consequently, the watermarking schemes can maintain acceptable bit accuracy when only a small portion of the watermarked text is replaced.

*Table 4.* Clean performance with 200 tokens for text generation task. PPL is the median perplexity.

| | | AUC ↑ | TPR @1% ↑ | Bit Accuracy ↑ | PPL ↓ | Extraction Time (s) ↓ |
|---|---|---|---|---|---|---|
| | Non-Watermarked | - | - | - | 8.016 | - |
| 24 Bit | StealthInk | 0.9780 | 0.8475 | 0.8747 | **7.141** | **0.01** |
| | MPAC | 0.9959 | 0.985 | 0.9437 | 10.19 | **0.01** |
| | Qu et al. | **1.0** | **1.0** | **0.9987** | 42.63 | 0.18 |
| | Fernandez et al. | 0.9433 | 0.655 | 0.61 | 9.46 | 113 |
| 36 Bit | StealthInk | 0.9709 | 0.7475 | 0.8327 | **7.135** | **0.01** |
| | MPAC | 0.9923 | 0.982 | 0.8927 | 9.376 | **0.01** |
| | Qu et al. | **0.9975** | **1.0** | **1.0** | 39.87 | 0.82 |
| | Fernandez et al. | - | - | - | - | 45100 (Estimated) |
| 48 Bit | StealthInk | 0.9556 | 0.6185 | 0.7867 | **6.968** | **0.01** |
| | MPAC | 0.9907 | 0.98 | 0.8611 | 10.11 | **0.01** |
| | Qu et al. | **0.9998** | **1.0** | **0.9775** | 41.16 | 1.37 |
| | Fernandez et al. | - | - | - | - | 1.85e9 (Estimated) |

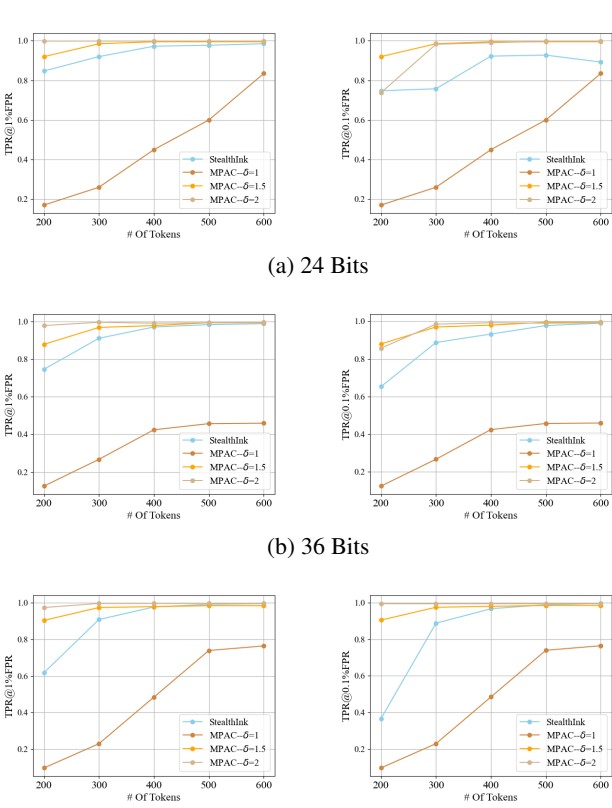

(a) 24 Bits

(b) 36 Bits

(c) 48 Bits

*Figure 5.* TPR of StealthInk and MPAC with number of tokens.

*Table 5.* Bit accuracy when launching copy-paste attack with 24 bits embedded in 300 tokens.

| Bit Accuracy | Copy-Paste Attack | | | |
|---|---|---|---|---|
| | $\epsilon=0$ | $\epsilon=0.1$ | $\epsilon=0.2$ | $\epsilon=0.3$ |
| StealthInk | 0.9195 | 0.8932 | 0.8649 | 0.8319 |
| MPAC | 0.9627 | 0.9572 | 0.9334 | 0.9075 |
| Qu et al. | 0.9987 | 0.9981 | 0.9993 | 0.9993 |
| Fernandez et al. | 0.6612 | 0.625 | 0.5762 | 0.5193 |

Table 6 shows that all evaluated schemes suffer significant performance degradation under paraphrasing attacks, with a marked drop in bit accuracy compared to the baseline ($\epsilon = 0$). This decline is due to the attack's alteration of sentence structure and rephrasing, which disrupts the embedded watermark signal. Achieving robustness against paraphrasing attacks remains a major challenge for multi-bit watermarking.

*Table 6.* Bit accuracy when using Dipper and PEGASUS to paraphrase 400 watermarked texts of 300 tokens with 24 bits embedded.

| Bit Accuracy | Paraphrasing Attack | | |
|---|---|---|---|
| | $\epsilon=0$ | $\epsilon=0.1$ | |
| | | Dipper | PEGASUS |
| StealthInk | 0.9195 | 0.6118 | 0.57 |
| MPAC | 0.9627 | 0.6108 | 0.5944 |
| Qu et al. | 0.9988 | 0.6337 | 0.6337 |
| Fernandez et al. | 0.6613 | 0.4162 | 0.4237 |

## 7. Conclusion

StealthInk is a provably stealthy and multi-bit watermarking scheme for LLMs that enables text traceability while preserving text quality. StealthInk uses a novel reweighting strategy for token probability distributions, ensuring unbiased distribution and effective multi-bit message embedding. Empirical results demonstrate StealthInk's high bit accuracy, superior text quality, and resilience against attacks. Our work highlights that watermarking must go beyond embedding and extraction — it must ensure statistical stealthiness to prevent detection and spoofing while maintaining text integrity. This balance enhances the practical applicability of watermarking in real-world scenarios. Future research should explore trade-offs among detectability, text quality, efficiency, and robustness to further strengthen watermarking for LLMs.

## Impact Statement

We introduce StealthInk, a multi-bit watermarking scheme for large language models (LLMs) that enables the stealthy embedding of meta-information into generated text. This allows an authorized user to verify the provenance of watermarked text without compromising its quality. Such a watermarking approach is crucial in addressing ethical concerns related to LLM misuse, including the creation of fake news, scams, and academic plagiarism. Recognizing these risks, major LLM providers—including OpenAI, Google, and Meta—have committed to incorporating watermarking techniques to mitigate misuse.

To assess the security of StealthInk, we evaluated its resilience against potential attacks aimed at forging the watermark or removing the watermark, including spoofing, copy-paste, and paraphrasing attacks. These attack vectors have been explored in prior watermarking research, and our focus was to compare StealthInk's robustness against other multi-bit LLM watermarking schemes. Through both theoretical analysis and empirical validation, we demonstrated StealthInk's advantages while also acknowledging its limitations.

LLM watermarking has the potential to serve as a valuable tool for preventing the ethical misuse of AI-generated text. Our work contributes to this effort by developing a stealthy, multi-bit watermark that balances security, detectability, and text quality. In summary, we believe this research does not raise any ethical concerns and instead provides a practical approach to improving AI accountability and trustworthiness.

## Acknowledgment

This research was partially supported by the Virginia Commonwealth Cyber Initiative (cyberinitiative.org) through the Cyber Acceleration, Translation, and Advanced Prototyping for University Linked Technology (CATAPULT) Fund.

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

# Appendix

# A. Related Work

### A.1. Zero-bit watermarking

Due to the discrete linguistic properties of text, watermarking for digital text is considered a challenging problem (Shih, 2017). Early approaches were mainly rule-based, such as paraphrasing (Atallah et al., 2002), syntactic structure restructuring (Atallah et al., 2001), and synonym substitution (Topkara et al., 2006). Later, advancements in modern language models led to improved methods. (Kirchenbauer et al., 2023) introduced the first watermarking scheme for LLMs and highlighted the critical property of the reweighting-based watermark by showing that the watermark could be detected algorithmically without any knowledge of the model parameters or access to the language model API. They categorized the vocabulary permutation into red and green lists using a hash function seeded with several previous tokens of the context text and proposed a reweighting strategy that simply adds a small bias to the logits of tokens in the green list. In this way, the tokens generated by the watermarked LLM will be biased to the green list. To detect the watermark, a detector possessing the hash function can recreate the lists and estimate the likelihood that the text is generated under reweighted probability distributions by hypothesis testing. Follow-up works proposed improvements to the robustness of this scheme against distortion-bounded attacks such as insertion, deletion, and substitution attacks (Kirchenbauer et al., 2024; Zhao et al., 2024).

By retaining the red-green list configuration, (Hu et al., 2024) and Dipmark (Wu et al., 2024) introduced an evolved family of permutation-based reweighting strategies for watermarking which maintains the expected distribution of the text; i.e., they proposed a stealthy or unbiased reweighting strategy for LLM watermarking. However, the detector in (Hu et al., 2024) necessitates access to both the prompt and the output distribution provided by the LLM for a given prompt, which requires the detector possesses knowledge of the prompt used to generate the detected text. In contrast, Dipmark requires no access to the prompts nor the LLMs. Nevertheless, the stealthiness of both (Hu et al., 2024) and Dipmark rely on independent permutations at each generation step, which means when encountering the same seed (i.e., several previously generated tokens) for pseudorandom permutation generation, the token is sampled without a watermark. In this way, as more and more queries are submitted to the LLM, the available seeds for watermarking decrease dramatically, and thereby the watermark strength is sacrificed for the stealthiness guarantee. Moreover, their approach is vulnerable to an effortless attacker who removes the watermark during text generation.

In contrast, to develop a stealthy watermark, (Christ et al., 2024) and (Kuditipudi et al., 2024) employed another line of sampling strategy without altering the probability distribution *at all* when generating the watermarked text, i.e., the inverse sampling method. However, resilience to text corruption in (Christ et al., 2024) is still an open problem. While the watermarking method of (Kuditipudi et al., 2024) was designed for robust detection, it requires secret key distribution during detection, which may potentially compromise the data security and stealthiness of the watermark. Moreover, the detection process in (Kuditipudi et al., 2024) involves hundreds of resampling steps from the secret key distribution, which is inefficient for lengthy texts. (Zhang et al., 2024b) trained the encoder-decoder modules to facilitate watermark insertion and extraction. However, the modules have to be learned for each LLM, which could be costly and inflexible. Besides, each text generated by the LLM will be processed by the encoder to embed the watermark, which will alter the original distribution of the LLM.

### A.2. Multi-bit watermarking

(Wang et al., 2024) extended (Kirchenbauer et al., 2023) but used the message content as the hash key before selecting the green list and further utilized a proxy language model for enhancing text quality. (Fernandez et al., 2023) proposed a technique for encoding a multi-bit message by providing a message-specific green list by shifting the vocabulary list dependent on the message. (Qu et al., 2024) built upon (Kirchenbauer et al., 2023), cyclically shifting vocabulary permutations according to the message and bias tokens in a green list to enable efficient multi-bit decoding. They also enhanced the embedding process by incorporating an error correction code into the message bits. However, overlapping shifts introduce interference, weakening message distinctiveness and statistical separation. To counteract this, a stronger bias is needed for reliable extraction, but this increases text distortion and reduces stealthiness. Crucially, these works achieved multi-bit watermarking by enumerating all potential messages or keys. Consequently, they involve computing across a space of potential messages that is exponentially large for the message length during decoding. This imposes a practical limit to the message length due to computational or memory constraints.

MPAC (Yoo et al., 2024) introduced a multi-color technique wherein the pseudorandomly generated vocabulary permutation,

with previously generated tokens as the seed, is divided into multiple equal-length segments and the segments are represented by a different color. Specifically, the message encoded into the text is formed by concatenating several message bits, each corresponding to a specific color segment. For instance, if the vocabulary is partitioned into four distinct colors, each message bit indicating the color interval to select should consist of two binary bits. In essence, if the message is composed of two message bits, then the message would be 4 binary bits long, as each position of the message bit is represented by two binary bits. During text generation, to embed the message using the next token, the index of the message bit is randomly chosen and the logits of the token in the color segment corresponding to the message bit are increased by a constant ($\delta$), which biases the selection of the next token to be sampled from that color interval.

However, these methods (Yoo et al., 2024; Qu et al., 2024) lack statistical stealthiness for potential attackers, making them very vulnerable to spoofing or scrubbing attacks, as we have evaluated. For example, MPAC is not stealthy according to Definition 4.1 (see Section 4), regardless of the value of $\delta$. To give an intuitive explanation for stealthiness, if we assume an attacker with knowledge of the output distribution of the original non-watermarked LLM can launch a sufficient number of queries for the same prompt and estimate the probability distribution of the first token, the attacker can easily determine the presence of a watermark in the generated text of the above works since the probability distribution of the first token reflected by the LLM is not the same as that of the original non-watermarked LLM. In fact, the attackers can distinguish the watermark and launch several advanced attacks such as watermark spoofing and scrubbing attacks (Jovanović et al., 2024) even without the knowledge of the original non-watermarked probability distribution.

In contrast to these approaches, which are unstealthy or rely on different keys to embed the messages, (Kordi Boroujeny et al., 2024) and (Zamir, 2024) developed multi-bit watermarking techniques in which the input and output distributions are the same, i.e., they are distortion-free. Nonetheless, since they build upon (Christ et al., 2024), the resilience of these multi-bit watermarking schemes remains an unresolved issue. On the other hand, AWT (Abdelnabi & Fritz, 2021) achieved unobtrusive encoding of text with a binary message through adversarial training of the encoder-decoder structure. However, AWT is prone to errors in message extraction and occasionally yields degraded watermarked texts due to its reliance on neural networks.

## B. Proof of Theorem 4.3

At the $k$th query attempt, for each prompt $\boldsymbol{a}^k$ and the response $\boldsymbol{x}^k$ of length $L^k$, to ensure $\boldsymbol{\theta}^k$ is independent, $\boldsymbol{\theta}_i^k$ ($i \in [1, L^k]$) cannot be applied to derive the reweighted probability distribution when the seed for PRF has been used during this query attempt. Instead, the original probability distribution $P_O(\boldsymbol{x}_i^k | \boldsymbol{a}^k, \boldsymbol{x}_{1:i-1}^k)$ should be used. Otherwise, the reweighted probability distribution $P_W^{M^k}(\boldsymbol{x}_i^k | \boldsymbol{a}^k, \boldsymbol{x}_{1:i-1}^k, \boldsymbol{\theta}_i^k)$ is applied to embed the watermark $M^k$.

$$\mathbb{E}_{\boldsymbol{\theta}^{1:K}, M^{1:K}} \left[ \prod_{k=1}^{K} \mathbb{P}_{wm}(\boldsymbol{x}^k | \boldsymbol{a}^k, M^k, \boldsymbol{\theta}^k, \boldsymbol{a}^{1:k-1}, \boldsymbol{x}^{1:k-1}, M^{1:k-1}) \right]$$

$$= \mathbb{E}_{\boldsymbol{\theta}^{1:K}, M^{1:K}} \left[ \prod_{k=1}^{K} \prod_{i=1}^{L^k} \mathbb{P}_{wm}(\boldsymbol{x}_i^k | \boldsymbol{a}^k, \boldsymbol{x}_{1:i-1}^k, M^k, \boldsymbol{\theta}^k, \boldsymbol{a}^{1:k-1}, \boldsymbol{x}^{1:k-1}, M^{1:k-1}) \right]$$

$$\overset{(a)}{=} \prod_{k=1}^{K} \mathbb{E}_{\boldsymbol{\theta}^k, M^k} \prod_{i=1}^{L^k} \mathbb{P}_{wm}(\boldsymbol{x}_i^k | \boldsymbol{a}^k, \boldsymbol{x}_{1:i-1}^k, M^k, \boldsymbol{\theta}^k, \boldsymbol{a}^{1:k-1}, \boldsymbol{x}^{1:k-1}, M^{1:k-1})$$

$$\overset{(b)}{=} \prod_{k=1}^{K} \mathbb{E}_{M^k} \prod_{i=1}^{L^k} \mathbb{E}_{\boldsymbol{\theta}_i^k} \mathbb{P}_{wm}(\boldsymbol{x}_i^k | \boldsymbol{a}^k, \boldsymbol{x}_{1:i-1}^k, M^k, \boldsymbol{\theta}^k, \boldsymbol{a}^{1:k-1}, \boldsymbol{x}^{1:k-1}, M^{1:k-1})$$

$$\overset{(c)}{=} \prod_{k=1}^{K} \mathbb{E}_{M^k} \prod_{i=1}^{L^k} P_O(\boldsymbol{x}_i^k | \boldsymbol{a}^k, \boldsymbol{x}_{1:i-1}^k, M^{1:k-1})$$

$$\overset{(d)}{=} \prod_{k=1}^{K} \mathbb{E}_{M^k} P_O(\boldsymbol{x}^k | \boldsymbol{a}^k)$$

$$= \prod_{k=1}^{K} P_O(\boldsymbol{x}^k | \boldsymbol{a}^k),$$

$$(8)$$

where (a) follows from independence of the sequence $\boldsymbol{\theta}^{1:K}$, (b) follows from the independence of $\theta_{1:L^k}^k$, (c) follows because either $P_W^{M^k}(\boldsymbol{x}_i^k | \boldsymbol{a}^k, \boldsymbol{x}_{1:i-1}^k)$ or $P_O(\boldsymbol{x}_i^k | \boldsymbol{a}^k, \boldsymbol{x}_{1:i-1}^k)$ is applied and from the unbiasedness of the reweighted probability distribution, and (d) follows from the chain rule for a joint distribution. The last equality follows from the randomness of the message, e.g., the timestamp will cause the message to be random.

## C. Proof of Theorem 4.4

Let $V = \{V_1, V_2, \ldots, V_{|V|}\}$ be the vocabulary of size $|V|$, and let $P_O = \{p_1, p_2, \ldots, p_{|V|}\}$ denote the original probability distribution over $V$. We define a permutation $\theta \sim \text{Unif}\{1, 2, \ldots, |V|!\}$. The cumulative probability for the top $t$ tokens under permutation $\theta$ is given by

$$X_t(\theta) = \sum_{j=1}^{t} p_{\theta(j)}. \tag{9}$$

With a slight abuse of notation, let token $V_i$ appear in position $t = \theta^{-1}(i)$ in the permutation. Then its position on the cumulative probability axis is the interval,

$$S_i = [x_i, x_i + p_i], \quad \text{where } x_i = \sum_{j=1}^{t-1} p_{\theta(j)}. \tag{10}$$

With $m$-bit message $M \in \{0, \ldots, 2^m - 1\}$ embedded, let $k = \frac{M|V|}{2^m}$ and $l = \frac{(M+1)|V|}{2^m}$. Then $\alpha(\theta)$ and $\beta(\theta)$ denote the permutation-dependent intervals,

$$\alpha(\theta) = \sum_{j=1}^{k} p_{\theta(j)}, \quad \beta(\theta) = \sum_{j=1}^{l} p_{\theta(j)}, \tag{11}$$

where $\alpha(\theta)$ is the sum of $k$ values sampled *without replacement* from $P_O$. By standard properties of sampling without replacement (Cohran, 1977) the expectation and variance of $\alpha$ are, respectively,

$$\mu_\alpha = \frac{k}{|V|} \quad \text{and} \quad \sigma_\alpha^2 = \frac{k(|V| - k)}{|V| - 1} \sigma^2, \tag{12}$$

where $\sigma^2 = \frac{1}{|V|} \sum_{i=1}^{|V|} \left(p_i - \frac{1}{|V|}\right)^2$ is the population variance of $P$.

When $|V|$ is sufficiently large, which is easily satisfied for LLMs (e.g., $|V| = 32,000$ for LLAMA2-7B), by the central limit theorem (CLT), $\alpha(\theta)$ can be approximated by a Gaussian distribution,

$$\alpha(\theta) \sim \mathcal{N}\left(\mu_\alpha, \sigma_\alpha^2\right), \tag{13}$$

and $x_i \sim \text{Unif}[0, 1 - p_i]$. Since

$$\beta(\theta) = \alpha(\theta) + \delta(\theta), \quad \text{where } \delta(\theta) = \sum_{j=k+1}^{l} p_{\theta(j)}, \tag{14}$$

Given $\alpha(\theta), \delta(\theta)$ is a sum over $r = \frac{|V|}{2^m}$ items sampled without replacement from the remaining $|V| - k$ tokens, which in total have mass $1 - \alpha(\theta)$. Then, under the CLT for sampling without replacement, we approximate

$$\delta \mid \alpha \sim \mathcal{N}\left(\mu_\delta(\alpha), \ \sigma_\delta^2(\alpha)\right), \tag{15}$$

where

$$\mu_\delta(\alpha) = \frac{r(1-\alpha)}{|V| - k}, \quad \sigma_\delta^2(\alpha) = \frac{r(|V| - k - r)}{|V| - k - 1} \sigma'^2, \tag{16}$$

where $\sigma'^2 = \frac{1}{|V|-k} \sum_{j=k+1}^{|V|} \left(p_{\theta(j)} - \frac{\sum_{j=k+1}^{l} p_{\theta(j)}}{|V|-k}\right)^2$. The value of $\sigma'^2$ depends on which tokens are selected into $\alpha$, and thus which tokens remain for $\delta$. However, since we do not know the specific indices $\theta(1), \ldots, \theta(k)$, we approximate the residual set of tokens as still following the original population distribution. Therefore, we assume $\sigma'^2 \approx \sigma^2$. In this way, $\sigma_\delta^2(\alpha) = \frac{r(d-k-r)}{(d-k-1)(d-k)} \sigma^2$.

Using the relationship $\beta = \alpha + \delta$, the conditional distribution of $\beta$ given $\alpha$ is

$$f_{\beta|\alpha}(\beta \mid \alpha) = \frac{1}{\sqrt{2\pi\sigma_\delta^2(\alpha)}} \exp\left(-\frac{(\beta - \alpha - \mu_\delta(\alpha))^2}{2\sigma_\delta^2(\alpha)}\right). \tag{17}$$

In this way,

$$(\alpha, \beta) \sim f_\alpha(\alpha) \cdot f_{\beta|\alpha}(\beta \mid \alpha), \tag{18}$$

and thus,

$$f_{\alpha,\beta}(\alpha, \beta) = \frac{1}{\sqrt{2\pi\sigma_\alpha^2}} \exp\left(-\frac{(\alpha - \mu_\alpha)^2}{2\sigma_\alpha^2}\right) \cdot \frac{1}{\sqrt{2\pi\sigma_\delta^2(\alpha)}} \exp\left(-\frac{(\beta - \alpha - \mu_\delta(\alpha))^2}{2\sigma_\delta^2(\alpha)}\right). \tag{19}$$

Denote $A(\theta)$ as the interval where the portion of token intersects with $A(\theta)$ will be zeroed out, $B(\theta)$ as the interval where the portion of token intersects with $B(\theta)$ will be doubled. Then, under the four different cases, $A(\theta)$ and $B(\theta)$ are, respectively,

$$\begin{cases} \text{Case 1 or 2:} & A(\theta) = [\alpha(\theta), \beta(\theta)], \quad B(\theta) = [1 - \beta(\theta), 1 - \alpha(\theta)], \\ \text{Case 3:} & A(\theta) = [\alpha(\theta), 1 - \beta(\theta)], \quad B(\theta) = [\beta(\theta), 1 - \alpha(\theta)], \\ \text{Case 4:} & A(\theta) = [1 - \alpha(\theta), \beta], \quad B(\theta) = [1 - \beta(\theta), \alpha(\theta)]. \end{cases} \tag{20}$$

Since the updated probability of a token depends on its position on the cumulative probability axis,

$$p_i^w(\theta) = \int_{x_i}^{x_i + p_i} w(x; \theta) \, dx, \tag{21}$$

where $w(x; \theta)$ is defined as

$$w(x; \theta) = \begin{cases} 0, & x \in A(\theta), \\ 2, & x \in B(\theta), \\ 1, & \text{otherwise.} \end{cases} \tag{22}$$

We denote the reweighted probability of token $V_i$ under permutation $\theta$ as

$$p_i^w(\theta) = p_i - \mu_A(\theta) + \mu_B(\theta), \tag{23}$$

where

$$\mu_A(\theta) = |S_i \cap A(\theta)|, \quad \mu_B(\theta) = |S_i \cap B(\theta)|. \tag{24}$$

We can calculate the expected reweighted probability under Case $c$ as

$$\mathbb{E}_\theta[p_i^w | \text{Case } c] = p_i + \mathbb{E}_\theta \left[ |S_i \cap B(\theta)| - |S_i \cap A(\theta)| \big| \text{Case } c \right]. \tag{25}$$

According to (20), in different cases,

$$|S_i \cap B(\theta)| - |S_i \cap A(\theta)| = \min\{x_i + p_i, B(\theta)_{[1]}\} - \max\{x_i, B(\theta)_{[0]}\} - \left( \min\{x_i + p_i, A(\theta)_{[1]}\} - \max\{x_i, A(\theta)_{[0]}\} \right) \tag{26}$$

where for example, in Case 1 or 2, $A(\theta)_{[0]} = \alpha(\theta)$ and $A(\theta)_{[1]} = \beta(\theta)$.

Above all, the expected reweighted probability is denoted as

$$\mathbb{E}_\theta[p_i^w] = \sum_{c=1}^{4} \mathbb{P}[\text{Case } c] \cdot \mathbb{E}_\theta[p_i^w | \text{Case } c], \tag{27}$$

and the difference between $\mathbb{E}_\theta[p_i^w]$ and $p_i$ is

$$\epsilon_i = \mathbb{E}_\theta[p_i^w] - p_i = \sum_{c=1}^{4} \mathbb{P}[\text{Case } c] \cdot \mathbb{E}_\theta[|S_i \cap B(\theta)| - |S_i \cap A(\theta)| \big| \text{Case } c]. \tag{28}$$

To derive $\epsilon_i$, we define $\Delta_i^c = E_\theta \left[ |S_i \cap B(\theta)| - |S_i \cap A(\theta)| \big| \text{Case } c \right]$. Therefore,

$$\Delta_i^c = \int_0^{1-p_i} \int_\alpha \int_\beta f_{\alpha,\beta}(\alpha, \beta) \cdot \frac{1}{1 - p_i} \cdot (|S_i \cap B(\theta)| - |S_i \cap A(\theta)|) \, d\beta \, d\alpha \, dx_i. \tag{29}$$

Let $S_i^{'} = [1 - x_i - p_i, 1 - x_i]$, which is symmetric to $S_i$. Since $A(\theta)$ and $B(\theta)$ are symmetric,

$$|S_i^{'} \cap B(\theta)| = |S_i \cap A(\theta)|, \quad |S_i^{'} \cap A(\theta)| = |S_i \cap B(\theta)|. \tag{30}$$

Therefore,

$$|S_i^{'} \cap B(\theta)| - |S_i^{'} \cap A(\theta)| = -(|S_i \cap B(\theta)| - |S_i \cap A(\theta)|). \tag{31}$$

Let $g(x_i) = |S_i \cap B(\theta)| - |S_i \cap A(\theta)|$, then $g(1 - x_i - p_i) = |S_i^{'} \cap B(\theta)| - |S_i^{'} \cap A(\theta)|$. Therefore, $g(x_i) + g(1 - x_i - p_i) = 0$, which means that $g(x_i)$ is antisymmetric about $\frac{1-p_i}{2}$. In this way, for each fixed pair $(\alpha, \beta)$, the integral of $\Delta_i^c$ in the range $x_i \in [0, 1 - p_i]$ is evaluated to zero, i.e., $\Delta_i^c = 0$. Therefore, $\epsilon_i = \sum_{c=1}^{4} \mathbb{P}[\text{Case } c] \cdot \Delta_i^c = 0$.

## D. Algorithm 2

---

**Algorithm 2** StealthInk decoder

---

**Input:** Text $x_{1:L}$, secret key sk, texture key length $h$, texture key history hist, permutation generation function $f$, threshold $z$, vocabulary size $|V|$, message set $\mathcal{M}$, the length $H$ of $MsgSeq$, unit $\gamma = 2^{-m}$

**Output:** False or (True, $MsgSeq$)

    Initiate $R_{pos}^M = 0$ for $M \in \mathcal{M}$ and $pos \in [0, 1, \ldots, H-1]$

    **for** $i = h+1, \ldots, L$ **do**

        Generate texture key $s_i$ based on $x_{-h:}$

        Set a value $pos \in [0, 1, \ldots, H-1]$ with seed $s_i$

        Generate permutation $\theta_i = f(\text{sk}, s_i)$

        **for** $M \in \mathcal{M}$ **do**

            Derive red list from eq. (4)

            **if** $x_i \in RL^M$ **then**

                $R_{pos}^M = R_{pos}^M + 1$

            **end if**

        **end for**

    **end for**

    Let $M_{pos}^* = \arg\min_{M \in \mathcal{M}} R_{pos}^M$, $MsgSeq[pos] = M^*$ for $pos \in [0, 1, \ldots, H-1]$

    Calculate $Z(MsgSeq, x_{1:L})$ from (6)

    **if** $Z(MsgSeq, x_{1:L}) \leq z$ **then**

        **return** (true, $MsgSeq$) **Else return** false

    **end if**

---

## E. Clarification on the Difference between StealthInk with $m = 1$ and Dipmark

Although we set up $m = 1$ for StealthInk, StealthInk is fundamentally different from Dipmark (Wu et al., 2024) in both design and functionality, and Dipmark cannot be adapted to embed a multi-bit watermark using the multi-chunk mechanism. In StealthInk, under $m=1$, if we embed bit 0 at a certain generation step, then $\alpha=0$ and $\beta$ is the cumulated probability of the tokens in the first half of vocabulary permutation (see (2) and (3)). Then at the detector, the watermarked token would not be sampled from the first half of the vocabulary permutation because its token probabilities are reweighted to zero. Therefore, the detector knows the exact permutation and the interval of each message, and can deterministically identify whether a token falls within the red list — enabling bit-accurate decoding. Each text generation step may have different $\alpha$ which depends on the embedded message. By contrast, $\alpha$ as denoted in Dipmark is fixed for each text generation step, which represents the probability interval in $[0, \alpha]$ will be reweighted to 0. In their detector, the secret key only determines a permutation of the vocabulary, and the detector must guess the green/red list separator $\gamma$ (e.g., 0.5) to compute a green-token ratio over the entire text. This design works well for zero-bit watermarking, but cannot recover message bits, nor can it guarantee bit accuracy when chunking the text to encode multiple bits. Therefore, chunking Dipmark to encode 1 bit per segment would compromise decoding accuracy, as it lacks a per-step message-aware reweighting function and cannot verify individual bit intervals without brute-force search over the message space.

While we fixed $m = 1$ in the main experiments for consistency and fair comparison with prior work, increasing $m$ does not necessarily degrade performance. In Fig. 4, we notice that based on our random selected prompts, $m = 1$ would be better for detectability than $m = 2$. However, as shown in our theoretical analysis in Fig. 3a, the bit-per-token rate can actually improve when $m$ increases, especially under high-entropy scenarios. For instance, under uniform distribution with maximum entropy and fixed eer of $0.01$, $m = 2$ achieves a higher bit-per-token rate than $m = 1$ (e.g., $2/42$ vs. $1/30$). The impact of $m$ is content-dependent, and our framework supports varying $m$ to trade off between capacity and detectability.

## F. Discussion on the Resilience of StealthInk against Adaptive Attacks

We assume there is an attacker who knows the probability distribution of an unwatermarked model and would like to infer the watermark by examining whether the distribution of responses generated by the watermarked model is the same as that of the unwatermarked model. If the attacker cannot infer the watermark, then the watermark must be stealthy. He/she could launch a large number of query attempts (e.g., 100,000 or more) for the same prompt to derive the watermarked distribution. However, using our method, since randomness is injected through the timestamp metadata in the watermark (milliseconds), even though the same permutation and unwatermarked probability distribution is produced for these attempts, different

messages would involve different reweighting functions. Therefore, each attempt will result in a distinct watermarked distribution of the watermarked text, including the first couple of tokens. Thus, the averaging of these probabilities renders the spoofing attack ineffective. Additionally, because of the stealthiness property, on average, the probability distribution of the first generated token (i.e., calculate the probability of each first token in responses) would be preserved as the probability distribution of the first token from the original unwatermarked model. If the attacker launches these queries at the exact same time (e.g., 10:30:51:02, 03/28/2025) using the same userID, model, etc., then the first tokens will be generated with the same message embedded and therefore from the same distribution. In this case, the attacker could infer that the probability distribution of the first token is distorted, i.e., a bunch of tokens' probabilities are 0 because the red tokens across these attempts are the same and will be reweighted to zero probability, which is different from the original distribution. However, launching such a large number of queries at the same exact time is practically impossible.

Furthermore, in the scenario where an attacker constructs a suitable prompt to make the first few tokens almost deterministic (e.g., using a problem template like "ANSWER:\n" to fix the first several tokens), the watermark can be embedded only after the prefix. Any deterministic tokens can not include any watermark. Hence, as the watermarking starts after any fixed tokens, StealthInk still satisfies $K$-shot stealthiness. The only problem is that the detector would also check the prefix when detecting the watermark in the given text. Since the prefix is non-watermarked, mixing it in the watermarked response could impact the detection performance. However, it is more like a copy-paste attack, which mixes a proportion of non-watermarked text into the watermarked text. We discussed it in Section 6.3, and in Table 5 it shows the bit accuracy will not be compromised significantly when the proportion of non-watermarked text is small.

## G. The Relationship between $l_{\min}$ and eer

We derive the relationship between the minimum length of text $l_{\min}$ for detection and the equal error rate (eer). For capacity $m$, we subdivide the interval $[0, 1]$, formed by tokens' cumulated probability according to the vocabulary permutation and original distribution, into subintervals of length $\gamma = 2^{-m}$. Then the red list of $\gamma_0$ is $[0, 2^{-m}]$, while the red list of $\gamma_{2^m-1}$ is $[1 - 2^{-m}, 1]$. Theoretically, for a non-watermarked text, the number of tokens in the red list of $\gamma_M$ follows a binomial distribution, $L_R^{\gamma_M} \sim \text{Bin}(L, \gamma)$, for all $M \in \mathcal{M}$. Since $\text{FPR} = \mathbb{P}\left[\min_{M \in \mathcal{M}} L_R^{\gamma_M} \leq \eta | L^{NW}\right] \geq \mathbb{P}[L_R^{\gamma_M} \leq \eta | L^{NW}]$ for all $M \in \mathcal{M}$, we can establish a lower bound for the false positive rate by setting

$$\Phi\left(\frac{\eta - L\gamma}{\sqrt{L\gamma(1 - \gamma)}}\right) = \text{FPR}$$
$$\implies \eta = \Phi^{-1}(\text{FPR})\sqrt{L\gamma(1 - \gamma)} + L\gamma, \tag{32}$$

where $\Phi(x)$ denotes the cumulative distribution function of the standard normal distribution.

On the other hand, for a watermarked text with $\gamma_{M^*}$ embedded, if $\alpha$ and $\beta$ are as in Case 1 or Case 2 in Fig. 2, no tokens will be sampled from the red list of $\gamma_{M^*}$, which we call the no-overlapping case ($\bar{\mathcal{O}}$). In contrast, under Case 3 or Case 4, tokens could be sampled from the red list of $\gamma_{M^*}$, which we call the overlapping case ($\mathcal{O}$). Obviously, $\mathbb{P}[\mathcal{O}] = 2^{-m}$ and $\mathbb{P}[\bar{\mathcal{O}}] = 1 - 2^{-m}$. When context n-grams are repeated, the next token will be sampled from $P_M$, which may result in the token falling in the watermarking region as well. Assuming that the context n-grams are repeated with probability $p$, denoted as $\mathbb{P}[\mathcal{R}] = p$, the probability of non-repetition is $\mathbb{P}[\bar{\mathcal{R}}] = 1 - p$.

Next, we derive the probability that the token falls in the red list of $\gamma_{M^*} \in \mathcal{M}$:

$$\mathbb{P}[rt^{\gamma_{M^*}}] = \mathbb{P}[\bar{\mathcal{R}}]\,\mathbb{P}[rt^{\gamma_{M^*}}|\bar{\mathcal{R}}] + \mathbb{P}[\mathcal{R}]\,\mathbb{P}[rt^{\gamma_{M^*}}|\mathcal{R}] \tag{33}$$

where

$$\mathbb{P}[rt^{\gamma_{M^*}}|\bar{\mathcal{R}}] = \mathbb{P}[\mathcal{O}]\,\mathbb{P}[rt^{\gamma_{M^*}}|\bar{\mathcal{R}}, \mathcal{O}] + \mathbb{P}[\bar{\mathcal{O}}]\,\mathbb{P}[rt^{\gamma_{M^*}}|\bar{\mathcal{R}}, \bar{\mathcal{O}}]$$
$$= 2^{-m}\left(\int_{\alpha}^{\frac{\alpha+\beta}{2}} 2\left(\frac{y}{\alpha + \beta}\right)\,dy + \int_{\frac{\alpha+\beta}{2}}^{\beta} 2\left(\frac{\beta - y}{\alpha + \beta}\right)\,dy\right)$$
$$= \frac{\beta - \alpha}{2^{m+1}}. \tag{34}$$

Therefore,

$$\mathbb{P}[rt^{\gamma_{M^*}}] = \frac{(1 - p)(\beta - \alpha)}{2^{m+1}} + p(\beta - \alpha). \tag{35}$$

The distribution of $L_R^{\gamma_{M^*}}$ is binomial, i.e., $L_R^{\gamma_{M^*}} \sim \text{Bin}(L, \mathbb{P}[rt^{\gamma_{M^*}}])$. Similarly, $L_R^{\gamma_{M'}} \sim \text{Bin}(L, \mathbb{P}[rt^{\gamma_{M'}}])$ and the probability that the token falls in the red list of $\gamma_{M'}$ ($M' \neq M^*$) is

$$\mathbb{P}[rt^{\gamma_{M'}}] = \frac{1 - \mathbb{P}[rt^{\gamma_{M^*}}]}{2^m - 1}. \tag{36}$$

The bound on the false negative rate is given by

$$\text{FNR} \geq \mathbb{P}\left[\min_{M \in \mathcal{M}} L_R^{\gamma_M} \geq \eta | L^W\right]. \tag{37}$$

The right-hand side of (37) can be approximated as follows:

$$\mathbb{P}\left[\min_{M \in \mathcal{M}} L_R^{\gamma_M} \geq \eta | L^W\right] \approx \mathbb{P}\left[L_R^{\gamma_{M^*}} \geq \eta | L^W\right] = 1 - \Phi\left[\frac{\eta - L\mathbb{P}[rt^{\gamma_{M^*}}]}{L\mathbb{P}[rt^{\gamma_{M^*}}](1 - L\mathbb{P}[rt^{\gamma_{M^*}}])}\right]. \tag{38}$$

To derive $l_{\min}$, we start from the estimate of $L = l_{\min} = l_0$ and derive $\eta$ using (32). Then using that $\eta$, we derive the lower bound for the false positive rate as shown in (37). If the lower bound is larger than FNR, we increase $l_{\min}$ by 1, where the process is iterated until the lower bound is smaller than FNR. In this way, we can derive the theoretical relationship between $l_{\min}$ and eer.

## H. Experimental Details

Following Dipmark, we evaluate StealthInk on three tasks: machine translation, text summarization, and text completion. All experiments are conducted on the Nvidia A100 GPU with 40 GB of memory. We use SHA-256 as the pseudorandom function and a 1024-bit random bit string as the secret key sk. The default temperature is 1.0 and the texture key length $h$ is 3. The multinomial sampling strategy is applied during text generation. For the machine translation task, we utilize the WMT'16 English (En) to Romanian (Ro) dataset, comprising 1,999 examples in the test set. We employ the Multilingual Bart (MBart) model (Liu et al., 2020) along with its official tokenizer. In the text summarization task, we use the test set from the CNN-DM corpus (Hermann et al., 2015), consisting of 11,490 examples on BART-large (Liu et al., 2020). We compare StealthInk with MPAC and the watermarking schemes of (Qu et al., 2024) and (Fernandez et al., 2023). In MPAC, the green list bias $\delta = 2$, and (Qu et al., 2024) set $\delta = 6$, while in (Fernandez et al., 2023), $\delta = 0.5$, which follow their original settings, respectively.

To evaluate text quality, we show the ROUGE score, the BLEU score, the BERTS score, and the perplexity of the text.

*ROUGE Score.* For the summarization task, we evaluate the effectiveness of summaries using the ROUGE score (Lin, 2004), which measures n-gram overlap to determine how well the generated summaries capture key content from the reference summaries.

*BLEU score.* For the machine translation task, we utilize the BLEU score (Papineni et al., 2002), which highlights the lexical similarity between machine-generated translations and human reference translations.

*BERTScore.* BERTScore (Zhang et al., 2019) evaluates the similarity between two sentences by summing the cosine similarities of their token embeddings. We use BERTScore-F1 to assess performance in both text summarization and machine translation tasks.

*Perplexity.* Perplexity quantifies how well a probability model predicts a given sample. Lower perplexity values indicate better predictive accuracy of the model.

To demonstrate the detectability performance of watermarking schemes, we evaluate them using metrics such as AUC, bit accuracy, TPR@1%FPR, and extraction time. AUC reflects the overall detection capability, while TPR@1%FPR represents the true positive rate when the false negative rate is below 1%. Bit accuracy measures the number of bits correctly extracted, and extraction time quantifies the time required to extract the watermarked message from a given text.

## I. Text quality comparison on text completion tasks

Fig. 6 compares the perplexity (PPL) of different watermarking schemes at a generation length of 200 tokens. As the payload increases from 24 to 48 bits, the impact on text quality becomes more evident. StealthInk consistently preserves the

lowest perplexity among all watermarked outputs, closely aligning with the non-watermarked baseline, which indicates minimal degradation in text quality. In contrast, MPAC introduces moderate increases in perplexity, while the (Qu et al., 2024) method exhibits substantial degradation, especially under higher payloads. At 48 bits, the (Fernandez et al., 2023) method performs better than that of (Qu et al., 2024) but still lags behind StealthInk.

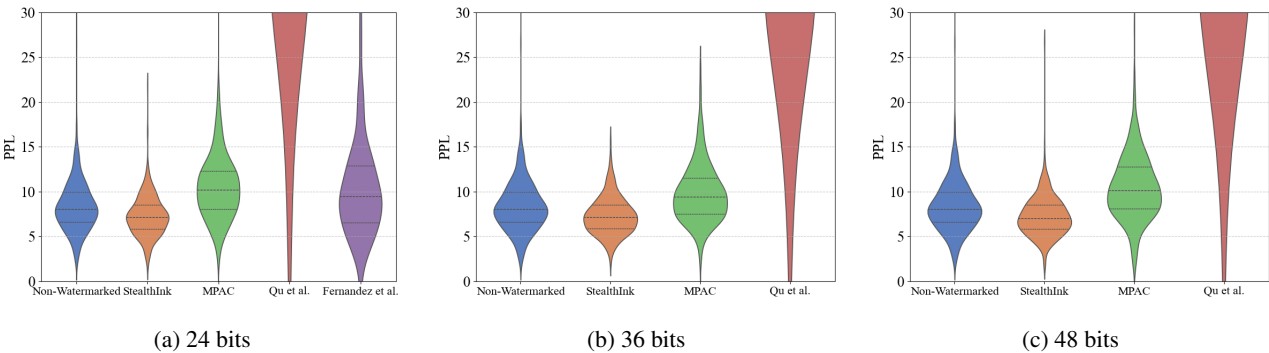

(a) 24 bits          (b) 36 bits          (c) 48 bits

*Figure 6.* Text quality of watermarking schemes at 200 tokens. For 36 and 48 bits, we omit (Fernandez et al., 2023) due to its high computational cost (see extraction time in Table 4).

## J. Explanation for lower PPL of StealthInk compared to non-watermarked sequences in Table 4

We show the PPL with various number of generations as in Table 7 and Table 8, which present the PPL statistics over responses of 200 tokens or 1000 tokens to 100 or 200 prompts. It is important to note that increasing the number of tokens per response leads to faster convergence in statistical properties compared to increasing the number of prompts, since the former reduces per-sample variance more effectively. As shown in the 1000-token results, the PPL gap is notably smaller across all watermark capacities, and the median values between non-watermarked and watermarked texts are very close.

Therefore, if we further increase the number of prompts or use even longer generated sequences, we expect the PPL statistics of StealthInk and non-watermarked text to converge more closely with a certain number of prompts or generated tokens per response. This aligns with the theoretical stealthiness guarantee, which holds in expectation over sufficient generations. The small residual gap in practice is a natural result of finite-sample variance rather than a violation of the stealthy design.

*Table 7.* PPL on responses to 100 prompts

| # of Tokens in Responses to 100 Prompts | 200 | | 1000 | |
|---|---|---|---|---|
| Statistics of PPL | Mean | Median | Mean | Median |
| Non-Watermark | 8.0437 | 8.0362 | 7.2706 | 7.3512 |
| StealthInk (24 Bit) | 7.4190 | 7.1819 | 7.1235 | 6.6913 |
| StealthInk (36 Bit) | 7.2576 | 7.1126 | 7.0746 | 7.0144 |
| StealthInk (48 Bit) | 7.1256 | 6.8348 | 6.9887 | 6.7862 |

*Table 8.* PPL on responses to 200 prompts

| # of Tokens in Responses to 200 Prompts | 200 | | 1000 | |
|---|---|---|---|---|
| Statistics of PPL | Mean | Median | Mean | Median |
| Non-Watermark | 8.1686 | 7.9392 | 7.2442 | 7.3142 |
| StealthInk (24 Bit) | 7.3745 | 7.2002 | 6.9904 | 6.7184 |
| StealthInk (36 Bit) | 7.2754 | 7.0635 | 7.0528 | 7.0834 |
| StealthInk (48 Bit) | 7.2118 | 6.9258 | 6.9232 | 6.6822 |

For additional clarification, please refer to Fig. 6. While the central body of the violin plots for StealthInk and non-watermarked text are similar—indicating comparable overall quality—some non-watermarked responses exhibit extremely high PPL (up to 30), which raises their median and average values. These outliers are not present in StealthInk-generated

text due to its filtering of low-probability tokens, which can incidentally lower PPL. If we instead evaluate PPL across a large number of responses to the same prompt, both StealthInk and non-watermarked text would converge to similar PPL values. That is because over many samples, StealthInk's reweighting does not introduce any overall bias into the output distribution; instead, it applies small, randomized modifications based on each specific prompt and embedded message to encode the watermark while maintaining stealthiness.

## K. Fig. 7: AUC and Bit Accuracy of StealthInk with Number of Tokens

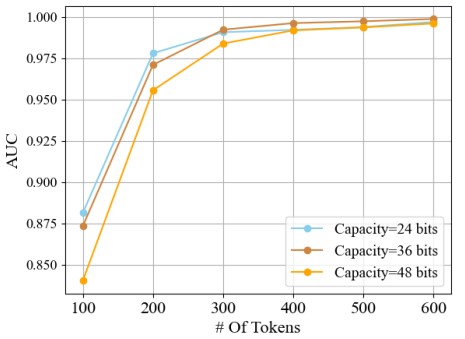 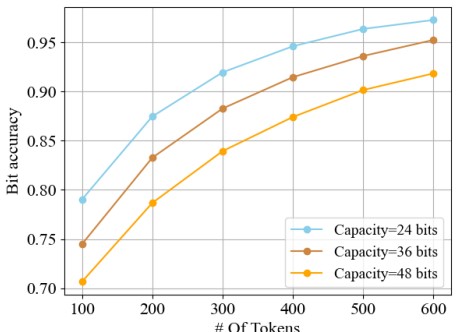

*Figure 7.* Performance of StealthInk across increasing number of tokens.

## L. Capacity Enhancement through Key Iteration

Intuitively, the watermark keys can be used to embed messages. For instance, to embed a 2-bit message, the watermark key can be selected from 4 possible candidates. Consequently, the embedding capacity can be improved along two dimensions simultaneously: first, by refining the watermarking scheme itself, such as with StealthInk, and second, by incorporating key iteration.

To evaluate this enhancement, we compare the performance of StealthInk with 24 bits embedded using its reweighting strategy and additional $m'$ bits embedded via key iteration over 200 tokens. As a result, a total of $(24 + m')$ bits are embedded across these two dimensions. See the details in Algorithm 3 and Algorithm 4.

---

**Algorithm 3** StealthInk Encoder Enhanced by Key Iteration

---

**Input:** Prompt $\boldsymbol{a}$, secret key set $\mathsf{SK} = \{\mathsf{sk}_0, \mathsf{sk}_1, \ldots, \mathsf{sk}_{2^{m'}-1}\}$, generation length $L$, texture key length $h$, permutation generation function $f$, unit $\gamma$, embedded message $[M_{key}, MsgSeq]$ where $M_{key} \in \{0, 1, \ldots, 2^{m'} - 1\}$ and $MsgSeq$ is a message sequence of length $H$, texture key history $\mathsf{hist} = \emptyset$.

  **for** $i = 1, \ldots, L$ **do**
    Calculate $P_O(\cdot\,|\boldsymbol{a}, \boldsymbol{x}_{:i-1})$
    Generate texture key $s_i$ based on $\boldsymbol{x}_{-h:}$
    Set a value $pos \in \{0, 1, \ldots, H - 1\}$ with seed $s_i$; let $M = MsgSeq[pos]$ and $\gamma_M = M\gamma$.
    **if** $s_i \in \mathsf{hist}$ **then**
      Sample $\boldsymbol{x}_i$ from $P_O(\cdot\,|\boldsymbol{a}, \boldsymbol{x}_{:i-1})$
    **else**
      $\mathsf{hist}.\mathsf{add}(s_i)$
      Generate permutation $\theta_i = f(\mathsf{sk}_{M_{key}}, s_i)$
      Calculate $\alpha$ and $\beta$ using (2) and (3)
      Calculate $P_W^M(\cdot\,|\boldsymbol{a}, \boldsymbol{x}_{:i-1}, \theta_i)$ from (5)
      Sample $\boldsymbol{x}_i$ from $P_W^M(\cdot\,|\boldsymbol{a}, \boldsymbol{x}_{:i-1}, \theta_i)$
    **end if**
  **end for**

---

Fig. 8 presents the AUC, TPR@1%FPR, and bit accuracy when enhancing the capacity of StealthInk along these two

---

**Algorithm 4** StealthInk decoder under Key Iteration

---

**Input:** Text $x_{1:L}$, secret key set $\mathsf{SK} = \{\mathsf{sk}_0, \mathsf{sk}_1, \ldots, \mathsf{sk}_{2^{m'}-1}\}$, texture key length $h$, texture key history hist $= \emptyset$, permutation generation function $f$, threshold $z$, vocabulary size $|V|$, message set $\mathcal{M}$, the length $H$ of $MsgSeq$, unit $\gamma = 2^{-m}$

**Output:** False or (True, $[M_{key}, MsgSeq]$)

  **for** $M_{key} \in \{0, 1, \ldots, 2^{m'} - 1\}$ **do**

    Initiate $R_{pos}^{\mathsf{sk}_{M_{key}}, M} = 0$ for $pos \in \{0, 1, \ldots, H - 1\}$ and $M \in \mathcal{M}$

    **for** $i = h + 1, \ldots, L$ **do**

      Generate texture key $s_i$ based on $x_{-h:}$

      **if** $s_i \in$ hist **then**

        Continue

      **end if**

      hist.$add(s_i)$

      Set a value $pos \in \{0, 1, \ldots, H - 1\}$ with seed $s_i$

      **for** $M \in \mathcal{M}$ **do**

        Generate permutation $\theta_i = f(\mathsf{sk}_{M_{key}}, s_i)$

        Denote red list $RL^M = \theta_i[\gamma_M |V| : (\gamma_M + \gamma)|V|]$

        **if** $x_i \in RL^M$ **then**

          $R_{pos}^{\mathsf{sk}_{M_{key}}, M} = R_{pos}^{\mathsf{sk}_{M_{key}}, M} + 1$

        **end if**

      **end for**

    **end for**

  **end for**

  Let $MsgSeq^{M_{key}}[pos] = \mathrm{argmin}_{M \in \mathcal{M}} R_{pos}^{\mathsf{sk}_{M_{key}}, M}$

  Calculate z-score $= Z(MsgSeq^{M_{key}^*}, x_{1:L})$ from eq.( 6) where $M_{key}^* = \arg\min_{\mathsf{sk}_{M_{key}} \in \mathsf{SK}} Z(MsgSeq^{M_{key}}, x_{1:L})$

  **if** z-score$\leq z$ **then**

    **return** (true, $[M_{key}^*, MsgSeq^{M_{key}^*}]$) **Else return** false

  **end if**

---

dimensions. The red dashed line represents the performance of StealthInk with 36 bits embedded at 200 tokens, without any capacity enhancement via key iteration. We observe that the performance without key iteration significantly outperforms the approach that enhances capacity through both dimensions. For instance, under the same number of tokens, the AUC of StealthInk with 36 bits embedded is 12% higher than that of StealthInk with 31 bits embedded, where $m' = 7$. Table 9 shows the detection time when detecting a text watermarked by StealthInk with $24 + m'$ bits. We can observe exponentially increased detection delay is introduced as $m'$ grows. Although the detection time for various watermark keys can be reduced by performing parallel detection across different watermark keys, the computational cost also increases as $m'$ grows, this additional overhead may limit the practicality of using multiple watermark keys.

Therefore, while key iteration can increase embedding capacity, it may introduce performance trade-offs, potentially reducing detectability and increasing the detection delay.

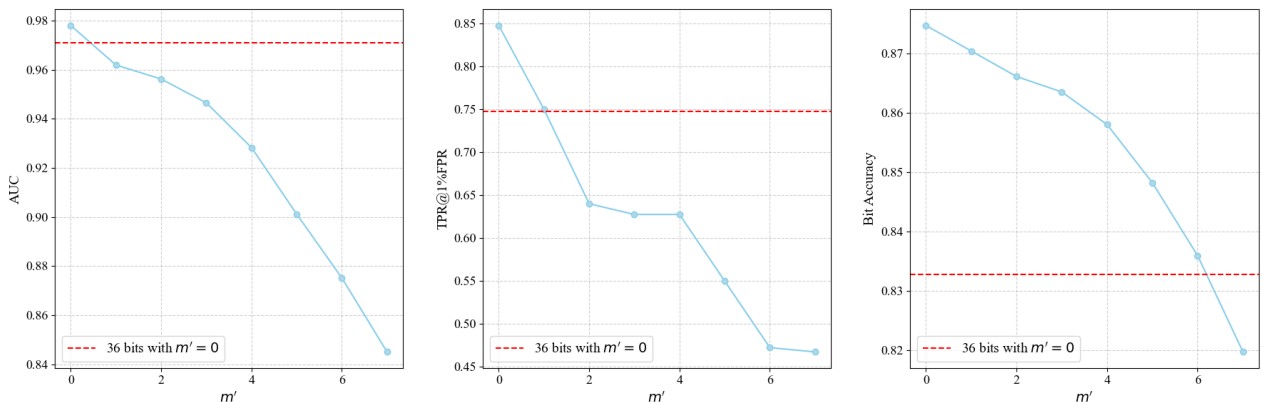

*Figure 8.* Performance when enhancing the capacity of StealthInk through two dimensions, with a total of $24 + m'$ bits embedded, where $m' \in [0, 1, 2, 3, 4, 5, 6]$.

*Table 9.* Dection time when enhancing the capacity of StealthInk through two dimensions, with totally $24 + m'$ bits embedded.

| $m'$ | 0 | 1 | 2 | 3 | 4 | 5 | 6 | 7 |
|---|---|---|---|---|---|---|---|---|
| Detection time (s) | 0.01 | 0.019 | 0.041 | 0.08 | 0.165 | 0.322 | 0.647 | 1.3 |

## M. Limitations and Future Works

Our approach offers valuable insights into the design of a stealthy and multi-bit watermarking system for LLMs; however, it does have some limitations. Firstly, our method is effective primarily for original outputs with high entropy. For generated text with low entropy, where there are a large number of tokens with high probabilities, our reweighting mechanism tends to apply Case 3 or Case 4 in Fig. 2. Consequently, some tokens within the watermarking interval may still be sampled, causing the text to be detected as non-watermarked. Secondly, while the stealthiness of our method makes it robust against spoofing attacks, it is equally important to develop a mechanism that can withstand sophisticated paraphrasing attacks, particularly given the advanced paraphrasing capabilities of modern LLMs. Moreover, as illustrated by (Zhang et al., 2024a), it is always possible to find a high-quality non-watermarked text that preserves the semantics of a watermarked text. However, a watermarking scheme that is robust against paraphrasing attacks can significantly increase the difficulty of identifying such a text. Future research should focus on strengthening watermarking techniques to further raise this barrier, making it more challenging for adversaries to remove or evade watermarks through paraphrasing. The spoofing attack of (Jovanović et al., 2024) targets zero-bit watermarking and may not easily extend to multi-bit schemes especially when the number of embedded bits increases. Thus, developing a more advanced spoofing attack for multi-bit watermarking and assessing the role of stealthiness is crucial.

## N. Watermarked Example

*Table 10.* Example of non-watermarked and watermarked texts generated by LLAMA-2-7B and StealthInk with 24 bits embedded at 200 tokens.

| |
|---|
| **Prompt:** Earthquake research has found that earthquakes are not evenly distributed across the earth, but are concentrated in certain specific strips. These strips where earthquakes are concentrated are called seismic belts. |
| **Message:** 111000110110110001010111 |
| **Non-watermarked:** |
| The north coast of the Pacific Ocean and the east coast of Africa, these are the well-known zones or strips where earthquakes occur. This makes you wonder if earthquakes are random in occurrence. No, they are not. They occur due to the interaction of the plates and the stress they are exerting on the Earth. 
 Seismic Hazard Areas or Zones 
 After studying the geographical regions where earthquakes occur more frequently, it was found that these places are in areas with volcanoes or subduction zones. However, not all the volcanoes or subduction zones are exactly the same. Some are more active, and there are fewer earthquakes that occur in these areas. However, there are many volcanoes that are rarely seen or felt by the people in that area. So, what is the difference between these volcanoes? Are there some volcanoes that are more active than others? |
| **Watermarked:** |
| In a seismic belt, one can observe a geographical zone where the earth's tectonic plates are interacting and forming tectonic boundaries. Also, these seismic belts or tectonic boundaries between the plates are usually found in areas where earthquakes occur. In these areas, there are usually complex subsurface structures such as volcanoes, subduction zones, and mountains. These structures and boundaries between plates generate stress that is observed by the people as earthquakes. There are many well-known seismic belts all over the world. However, most of the seismic belts or zones are found in places such as near the mid-ocean ridges. The mid-ocean ridges are located near the edges of the plates. Therefore, they contain seismic belts that separate plates from each other. The seismic belts are found in places such as |
| **Extracted Message:** 11100011011011000101011 |
| **Bit Accuracy:** 0.96 |
| **Non-watermarked text P Value:** 0.0006364708785215253 |
| **Watermarked text P Value:** 1.0457118616738534e-09 |

