# OpenReview forum: "StealthInk: A Multi-bit and Stealthy Watermark for Large Language Models"
_ICML.cc/2025/Conference — ICML 2025 poster_

### Official Review · Reviewer_DZ3i · 2025-03-05

**Overall Recommendation:** 2

**Summary:**

This paper proposes a multi-bit stealthy (a.k.a. unbiased) LLM watermark. The method is based on partitioning the text into different intervals, increasing the probabilities for some parts while decreasing the others, and keeping the overall distribution unchanged. The evaluation shows that the method can indeed have a better stealthiness than existing methods, but the detection performance is degraded.

**Claims And Evidence:**

The main claim of the paper is the stealthy multi-bit LLM watermark method. The stealthiness is supported by the theoretical proof of Theorem 2; the multi-bit property holds true by the design of the method.

**Essential References Not Discussed:**

Essential references are discussed.

**Experimental Designs Or Analyses:**

The experiment results do not look good to me. The most important metric of a LLM watermark, i.e. the detectability, is shown in Table 4 where the proposed method has lower performance compared to related works (e.g. MPAC). I understand that the better normal utility of the watermarked model is one advantage of the method, but the performance gap is not negligible - a AUC decrease from >0.99 to <0.98 is a big decrease, and if we evaluate through metrics like TPR@0.1%FPR, the gap will be more obvious.

**Methods And Evaluation Criteria:**

The paper uses the model normal utility (e.g. BERTScore/BLEU/PPL) to evaluate stealthiness and use binary classification metrics (AUC, TPR, Bit acc) to evaluate the detectability. I am concerned with the stealthiness metric, as a watermark can have a high normal utility while still have a bad stealthiness. That is, a watermark with distributional shift can still have a high BERTScore/BLEU/PPL on machine translation tasks. Therefore, I do not think the normal utility is a good indicator of stealthiness. For detectability tasks, the metrics are generally good but TPR@10%FPR is not a good fit - it is rare to tolerate 10% FPR for most tasks in watermarks. Metrics like TPR@0.01%FPR would be a better measure.

**Other Comments Or Suggestions:**

The template is broken - headers and author lists are missing.

**Other Strengths And Weaknesses:**

The strengths and weaknesses are stated in the sections above.

**Questions For Authors:**

How does the detectability compare with other methods when evaluated with TPR@FPR=1% and TPR@FPR=0.1%?

**Relation To Broader Scientific Literature:**

This paper proposes a new method on the line of multi-bit LLM watermark research. The idea of increasing part of the probability distribution and decreasing the others could be an interesting finding to the field.

**Theoretical Claims:**

I did not check the mathematical details of the proofs for Theorem 1&2, but they make intuitive sense to me.

---

> ### Author Rebuttal · Authors · 2025-04-01
>
> # "Methods And Evaluation Criteria" and "Relation To Broader Scientific Literature" and "Questions For Authors":
>
> We appreciate the reviewer’s insightful observation. We fully agree that normal utility metrics (e.g., BLEU, BERTScore, PPL) alone are not sufficient to evaluate stealthiness, as they may not reveal subtle distributional shifts. However, we would like to clarify the purpose and precedent for using these metrics in our work.
>
> Our paper does not treat utility metrics as indicators of stealthiness, but rather uses them to verify that text quality is preserved — a necessary condition for stealthy watermarks, though not sufficient. We rigorously define and prove stealthiness via distributional indistinguishability in Section 4.1 and Theorem 2, and we evaluate detectability empirically through spoofing attacks, false positive rates, and hypothesis testing (e.g., Table 3).
>
> Importantly, our zero-bit watermarking baseline DiPmark — which is specifically designed to be distribution-preserving — also follows the same evaluation strategy. As shown in the DiPmark paper (Figure 2 and Table 3), the authors explicitly use BLEU and PPL to demonstrate that DiPmark preserves the quality and distribution of the original language model output. These metrics are used to contrast DiPmark against distribution-modifying methods (e.g., Kirchenbauer et al., 2023), which exhibit clear utility degradation. Thus, reporting these metrics is a widely accepted practice for verifying that watermarking mechanisms do not compromise generation quality.
>
> Regarding the metrics of TPR@ 1\%FPR and TPR@ 0.1\%FPR: We thank the reviewer for the valuable feedback. In response, we have extended our evaluation to include TPR@ 0.1\%FPR and TPR@ 1\%FPR across different token lengths, which we agree are highly relevant for understanding watermark detectability under low false positive constraints. [View TPR@ 0.1\%FPR and TPR@ 1\%FPR comparation](https://github.com/AnonymousLink123/anonymousREBUTTAL/blob/main/Figures%20of%20TPR.pdf) through this anonymous link.
>
> We found that while StealthInk may initially show slightly lower TPR at shorter lengths (e.g., 200 tokens), its detectability approaches or matches MPAC as the number of tokens increases. For example, when embedding 36 bits, MPAC ($\delta$=2, i.e., original setting in MPAC) achieves TPR@ 1\%FPR of 0.98 at 200 tokens, while StealthInk achieves comparable TPR@ 1\%FPR (i.e., 0.985) at 400 tokens. Besides, we also compare the performance of StealthInk with MPAC ($\delta$=1) and MPAC ($\delta$=1.5) in the figures. We can observe that StealthInk is significantly better than MPAC ($\delta$=1), while close to MPAC ($\delta$=1.5). For example, when embedding 36 bits, MPAC ($\delta$=1.5) achieves TPR@ 1\%FPR of 0.97 at 300 tokens, while StealthInk achieves comparable TPR@ 1\%FPR (i.e., 0.9725) at 400 tokens. Besides, StealthInk achieves better TPR@ 0.1\%FPR than MPAC ($\delta$=1.5) across different number of tokens.
>
> This demonstrates that StealthInk is competitive in detectability given sufficient sequence length, while offering additional benefits in terms of stealthiness, robustness to spoofing, and multi-bit capacity. We would also like to emphasize that our design intentionally trades off a small degree of detectability in favor of stronger stealthiness guarantees. Unlike distribution-modifying schemes like MPAC, which achieve higher detectability through aggressive token-level distortion, StealthInk is designed to be statistically stealthy across multiple generations with theoretical guarantees.
>
> We will add two columns of TPR@ 1\%FPR and TPR@ 0.1\%FPR in Table 4 and include these additional results and analysis in the appendix of revised manuscript to provide a more complete picture of the detectability-performance trade-off over varying token lengths.
>
> # Other Comments Or Suggestions:
>
> We thank the reviewer for pointing this out. We will correct it in the revision.

---

### Official Review · Reviewer_BuPL · 2025-03-08

**Overall Recommendation:** 3

**Summary:**

The paper introduces StealthInk, a watermarking scheme for LLMs that embeds multi-bit information into AI-generated text without disrupting the original text distribution. Unlike previous methods that either altered text outputs or limited watermarks to simple detection, StealthInk preserves the generative quality of LLMs while adding traceable data. The authors develop a mathematical framework to establish a lower bound on the token count needed for reliable watermark detection at a predetermined equal error rate, optimizing the scheme’s capacity for different use cases.

**Claims And Evidence:**

All claims made in the submission supported by clear and convincing evidence.

**Essential References Not Discussed:**

N/A

**Experimental Designs Or Analyses:**

I checked all experimental designs and analyses. The experimental settings are generally the same as prior work.

**Methods And Evaluation Criteria:**

There is a flaw in the proposed method. In the proposed method, the authors use two parameters m and H to embeds m*H bits signal into the content. However, in experiments, the authors fix m=1, because increasing m will hurt the performance. With m=1, we will have beta=1, and the proposed reweight become the reweight used in dipmark. Thus, the method used in the experiment is generally dipmark plus a multi-chunk mechanism for embedding multi-bit information, which hurts the originality of this work.


Most of the evaluation criteria make sense. However, for measuring the detectability, the authors only report TPR@10% FPR, which is not a practical metrics. Usually we consider TPR@1% FPR and TPR@0.1% FPR (Kirchenbauer et al., 2023).

**Other Comments Or Suggestions:**

It seems that the authors did not use the ICML 2025 template provided on the official website.

**Other Strengths And Weaknesses:**

N/A

**Questions For Authors:**

In Table 4, the PPL of the StealthInk are significantly lower than the non-watermarked sequences, which contradicts to the steathy property of StealthInk. Can the authors explain the possible reason for this observation?

**Relation To Broader Scientific Literature:**

From my perspective, the key contribution of this paper is the stealthy reweight method, which is developed upon dipmark. If we set m=1, the method in Figure 1 is just the same as dipmark.

**Theoretical Claims:**

No.

---

> ### Author Rebuttal · Authors · 2025-04-01
>
> # "Methods And Evaluation Criteria" and "Relation To Broader Scientific Literature":
> We thank the reviewer for the thoughtful comments. We clarify several key points regarding the originality of our work relative to DiPmark.
>
> Although we set $m=1$ in our main experiments, StealthInk is fundamentally different from DiPmark in both design and functionality. In StealthInk, the reweighting strategy is message-dependent at each generation step. For example, embedding bit 0 results in $\alpha=0$ and $\beta$ as the cumulative probability of tokens in the first half of the permutation (see Eq. (1) and (2)). During detection, the token will not fall within the red list, whose probabilities are zeroed, enabling bit-accurate decoding without ambiguity. By contrast, $\alpha$ as denoted in Dipmark is fixed for each generation step, which represents the probability interval in [0, $\alpha$] will be reweighted to 0. In their detector, though a vocabulary permutation for each token can be reproduced using the secret key, the detector must guess the green/red list separator $\gamma$ (e.g., 0.5) to compute a green-token ratio over the entire text. This design works well for zero-bit watermarking, but cannot recover message bits, nor can it guarantee bit accuracy when chunking the text to encode multiple bits. Therefore, chunking DiPmark to encode 1 bit per segment would compromise bit accuracy, as it lacks a per-step message-aware reweight function and cannot verify individual bit intervals.
>
> On the claim that increasing $m$ hurts performance: While we fixed $m=1$ in our main experiments, increasing $m$ does not inherently degrade performance. The effect depends on the entropy of the red tokens in the vocabulary permutation, which relates to the interval size $\beta - \alpha$ in Eq. (6). As shown in Figure 4, $m=1$ performs better on our selected prompts, but our theoretical analysis in Figure 3(a) shows that higher $m$ can improve bit-per-token rate—especially in high-entropy settings. For example, under a uniform distribution (i.e., maximum entropy) with EER of 0.01, $m=2$ achieves 2/42 bits per token v.s. 1/30 for $m=1$. Thus, $m$ offers a tunable trade-off between capacity and detectability, depending on the content.
>
> Regarding the metrics TPR@ 1%FPR and TPR@ 0.1%FPR: Due to the limit space, please refer to our response to the reviewer DZ3i about the evaluation metrics under two FPR and analysis.
>
> # Questions For Authors:
>
> We appreciate the reviewer’s thoughtful observation. The difference in perplexity (PPL) values arises from the evaluation setup rather than a violation of StealthInk’s stealthiness.
>
> StealthInk applies two constraints during sampling: (1) it removes tokens in a red list by zeroing their probability, and (2) samples from the remaining tokens via multinomial sampling. In contrast, non-watermarked generation only applies the second step. In low-entropy scenarios, the red list mostly removes unlikely, low-quality tokens, slightly improving PPL. In high-entropy scenarios, it may exclude moderate-probability tokens, but sampling still favors higher-probability ones. In both cases, StealthInk reduces the chance of selecting semantically weak tokens, which can lead to slightly lower PPL.
>
> However, this does not contradict the stealthiness of StealthInk. The PPLs in Table 4 are aggregated over 200-token responses to 500 prompts, each with a random message and permutation. Natural variability across prompts and messages can cause small empirical differences. However, StealthInk is provably stealthy in expectation (see Definition 1 and Theorem 2): over many samples or longer outputs, the token distributions of watermarked and non-watermarked text converge.
>
> To support this, we include PPL statistics for generations of 200 and 1000 tokens across 100 and 200 prompts in the anonymous link [View the PPL statistics](https://github.com/AnonymousLink123/anonymousREBUTTAL/blob/main/PPL%20Statistics.png). As shown, increasing the number of tokens per response significantly reduces the PPL gap. With 1000-token generations, the PPL gap is notably smaller, and median values are nearly identical across watermark capacities. Thus, if we further increase the number of prompts or sequence length, we expect even tighter convergence between the two distributions—consistent with StealthInk’s theoretical guarantees. The small remaining differences are due to finite-sample effects, not a flaw in the method. Figure 5 in Appendix G further illustrates this. The violin plots show overall similarity, with the non-watermarked PPL inflated by a few high-outlier responses (up to 30), which are less likely under StealthInk due to red list filtering. Evaluating many samples for the same prompt would yield even closer PPL values.
>
> We will clarify this distinction in the revision to avoid confusion.
>
> # Other Comments Or Suggestions:
>
> We thank the reviewer for pointing this out. We will correct it in the revision.

---

### Official Review · Reviewer_KJSb · 2025-03-14

**Overall Recommendation:** 4

**Summary:**

This paper introduces a novel watermarking scheme that allows for the stealthy embedding of multi-bit information within generated text. This method aims to enhance the traceability of AI-generated content while preserving the original text quality and ensuring robustness against various attacks.

**Claims And Evidence:**

The paper makes several claims regarding its watermarking scheme, and it generally provides substantial evidence to support these claims.

**Essential References Not Discussed:**

No

**Experimental Designs Or Analyses:**

The authors have conducted a comprehensive set of experiments to validate their claims.

**Methods And Evaluation Criteria:**

The time complexity for extracting the multi-bit information is quite impressive as it does not increase as the size of the information increases.

**Other Comments Or Suggestions:**

No

**Other Strengths And Weaknesses:**

The paper provides a theoretical derivation of the minimum number of tokens required for watermark detection at a fixed equal error rate.
The authors formally define stealthiness for multi-bit watermarking, extending the concept from zero-bit watermarking.
The evaluation is comprehensive as detection accuracy, robustness, speed and text quality are all well covered.

**Questions For Authors:**

Please refer to previous comments

**Relation To Broader Scientific Literature:**

The paper proposed a watermark algorithm that achieves multi-bit capacity, robustness, efficiency and undetectability while previous works failed to achieve all.

**Theoretical Claims:**

The theoretical claims looks correct with proofs provided. No issues.

---

> ### Author Rebuttal · Authors · 2025-04-01
>
> We sincerely thank the reviewer for the positive evaluation of our work. We are glad to hear that our contributions were well received, including the theoretical foundation for multi-bit watermarking, the efficient and accurate decoding scheme, and the comprehensive evaluation.
>
> We particularly appreciate your recognition of our formal definition of stealthiness for multi-bit watermarking and the theoretical derivation of the minimum number of tokens required for detection at a fixed equal error rate. We are also encouraged that the reviewer found our method to achieve a great tradeoff between multi-bit capacity, stealthiness, robustness, and efficiency, which addresses limitations in prior work.
>
> Thank you again for your review and recommendation.

---

### Official Review · Reviewer_Js5V · 2025-03-28

**Overall Recommendation:** 3

**Summary:**

The paper proposes a novel multi-bit watermarking scheme, StealthInk, for large language models (LLMs). It discusses both the embedding and detection of watermarks, with theoretical and experimental support.

**Claims And Evidence:**

The main challenge addressed is multi-bit watermarking and authors' main claim is that their new method can solve this challenge. I find that the method definition and derivation are suitable for the multi-bit watermarking problem, with both theoretical proofs and experimental results supporting the claim, as detailed below.

**Essential References Not Discussed:**

No

**Experimental Designs Or Analyses:**

The paper uses appropriate evaluation metrics to assess the performance of watermarking method.

**Methods And Evaluation Criteria:**

The proposed method demonstrates significant novelty, introducing a new reweighting strategy designed specifically for multi-bit watermarking. The use of MPAC for position encoding is coming from Yoo et al., 2023, and I think it is a reasonable choice.

**Other Comments Or Suggestions:**

Regarding the handling of the history log in the paper, there are various methods in practice that can weaken the strict K-shot stealthiness to achieve a trade-off between practicality and stealthiness. The paper's approach is reasonable, but I didn't quite follow the discussion of an attack cost of 100,000. My understanding is that the paper's method injects randomness into the first token (e.g., milliseconds), but an attacker can always construct a suitable prompt to make the first few tokens almost deterministic (e.g., using a problem template like "ANSWER:\n" to fix the first two tokens). This would invalidate the randomness injection in the first token.

**Other Strengths And Weaknesses:**

The definitions of G are somewhat mixed together with F in eq (4). A clearer presentation would be beneficial.

**Questions For Authors:**

Please refer to the concerns raised in the "Theoretical Claims" section.

---

Update: I just noticed my separate comment is not visible to authors.

I want to thank authors for update.

I confirm the new $F_k$ is monotonically increasing for k and generates valid watermarked probability. However, I still don't catch all the proof. For $F\_{|V|+1-t}(\theta\_i^r,M,P\_O)=(X\_{|V|+1-t}^r-\beta)^++(X\_{|V|+1-t}^r-\bar{\beta})^+-(X\_{|V|+1-t}^r-\alpha)^--(X\_{|V|+1-t}^r-\bar{\alpha})^+$, does $\alpha$ and $\beta$ depend on $\theta$? When $\theta$ is reverse to $\theta^r$, do we have a new $\alpha^r$ and $\beta^r$?

**Relation To Broader Scientific Literature:**

This paper makes a novel contribution to the field of multi-bit watermarking for large language models.

**Theoretical Claims:**

The overall theoretical flow is natural and appropriate, starting with the definition of stealthy or unbiased multi-bit watermarking, inducing the scheme, proving unbiasedness, and deriving the detection method. However, there are some concerns:
1. Eq (4) may have a typo, as the expression for Case 2 appears to be the same as Case 1, which seems incorrect.
2. The proof of Theorem 2 may have a mistake, as line 771 does not correspond to any case in eq (4).
3. I am not entirely confident in understanding the proof of Theorem 2. I would like to have confirmation from the authors regarding: When reversing $\theta$ to $\theta^r$, should the message $M$ also be reversed to $M^r$? In other words , is it necessary to perform an additional shuffle of M based on $s_i$ at each step to ensure interval symmetry? I may not have fully grasped the proof and wish to communicate with the authors to understand these points. I am willing to adjust score once I confirmed these details to be correct.

---

> ### Author Rebuttal · Authors · 2025-04-01
>
> # Theoretical claims:
>
> We correct eq. (4) as
> $$
> F_{k}(\theta, M, P_{O}) =
> \begin{cases}
> (X_k - \beta)^+ + (X_k - \bar{\beta})^+ - (X_k - \alpha)^- - (X_k - \bar{\alpha})^+, & \text{Case 1 or 3} \\\\
> (X_k - \beta)^+ + (X_k - \bar{\beta})^- - (X_k - \alpha)^- - (X_k - \bar{\alpha})^-, & \text{Case 2 or 4}
> \end{cases}
> $$
> This is a typo in the paper. However, **we truly execute reweighting function as the corrected eq. (4)** in the experiments, which is indicated in lines 220-245 of previously submitted codes in the supplementary materials.
>
> Next, we provide the proof for theorem 2 in the anonymous link [View the anonymous proof (PDF)](https://github.com/AnonymousLink123/anonymousREBUTTAL/blob/main/Proof%20of%20Theorem%202.pdf). During the proof, when reversing $\theta$ to $\theta^{r}$, the message $M$ should not be reversed to $M^{r}$. This is crucial because the expectation is taken over permutations sampled from the uniform set $\Theta$, with a fixed message $M$. The reason is illustrated through the two terms that are marked as red and blue in the proof. In both the red and blue terms, we fix the message $M$ and vary the permutations across $\Theta$. For each $\theta$, there exists a corresponding reversed permutation $\theta^r \in \Theta$. If we were to reverse $M$ in the blue term, we would no longer be computing an expectation over the same distribution, because each $P^{M}_{W}(\cdot|\cdot, \theta)$ would no longer match its counterpart under reversal. The reweighted probabilities under $\theta$ and $\theta^r$ indeed cancel out asymmetries in the distribution only when $M$ is fixed, so the expectation over both directions is balanced.
>
> # Other Strengths And Weaknesses:
>
> We define $F_{k}(\theta, M, P_{O})$ as the cumulative function over the reweighted vocabulary permutation up to the $k$-th token. To recover the actual probability for each token, we define a differencing operator $G$ such that the reweighted probability for $k$-th token is: $$P^M_W(t_k | a, x_{1:i−1}, θ_i) = G(F_{k,k−1}(θ, M, P_O^i))=F_k(θ_i, M, P_O^i) − F_{k−1}(θ_i, M, P_O^i)$$ where $$F_0(\theta_i, M, P_O^i) = 0$$
>
> # Other Comments Or Suggestions:
>
> First, we illustrate the discussion of an attack cost of 100, 000 (line 241 in page 5). We assume there is an attacker who knows the probability distribution of unwatermarked model, and would like to infer the watermark by examining whether the distribution of responses generated by watermarked model is the same as unwatermarked model. If even the attacker cannot infer the watermark, then the watermark must be stealthy. He/she could do lots of query attempts (e.g., 100, 000 or more) for the same prompt to derive the watermarked distribution. However, using our method, since the randomness (milliseconds) is injected, even though the same permutation and unwatermarked probability distribution is produced for these attempts, different message would involve various reweighting functions. Therefore, each attempt will result in distinct watermarked distribution of the watermarked  text, including the first couple of tokens. Therefore, the averaging of these probabilities renders the spoofing attack ineffective. Additionally, because of stealthness property on average, the probability distribution of the first generated token (i.e., calculate the probability of each first token in responses) would be preserved as the probability distribution of the first token from the original unwatermarked model. Only if the attacker launches these queries at the exact same time (e.g., 10:30:51:02, 03/28/2024) using the same userID, model, etc., then the first tokens will be generated with the same message embedded and therefore from the same distribution. In this case, the attacker could infer the probability distribution of the first token is distorted, i.e., a bunch of tokens' probabilities are 0 because the red tokens across these attempts are the same and will be reweighted to zero probability, which is different from the original distribution. However, such large number of queries at the same exact time is practically impossible.
>
> Next, we answer the second concern of the reviewer from prefix-specifible LLM. Although ''Answer:'' is a prefix specified by the user, the watermark can be embedded only after the prefix. Any deterministic tokens can not include any watermark. Hence, as the watermarking starts after any fixed tokens, our watermarking method still satisfies $K$-shot stealthiness. The only problem is that the detector would also check the prefix when detecting the watermark in the given text. Since the prefix is non-watermarked, mixing it in the watermarked response could impact the detection performance. However, it is more like copy-paste attack, which mixes a proportion of non-watermarked text into the watermarked text. We discussed it in Section 6.3, and in Table 5 it shows the bit accuracy will not be compromised significantly when the proportion of non-watermarked text is small.

---

### Decision · Program_Chairs · 2025-05-01

**Decision:**

Accept (poster)

**Comment:**

The paper proposes StealthInk, a watermarking scheme for LLMs that embeds multi-bit information into AI-generated text without disrupting the original text distribution.  The authors develop a mathematical framework to establish a lower bound on the token count needed for reliable watermark detection at a predetermined equal error rate, optimizing the scheme’s capacity for different use cases.

Strengths:
- The proposed method is novel for multi-bit watermarking.
- The overall theoretical flow is natural and appropriate, starting with the definition of stealthy or unbiased multi-bit watermarking, inducing the scheme, proving unbiasedness, and deriving the detection method.
- The paper uses appropriate evaluation metrics to assess the performance of watermarking method.

Weakness:
- The paper only uses the setting for m=1, which limits the generality.
- There is some concern regarding performance drop in AUC (from 0.99 to 0.98)